# Shortcut Diffusion Training With Cumulative Consistency Loss: An Optimal Control View

**Paribesh Regmi**[1]       **Sandesh Ghimire**      **Rui Li**[1*]
pr8537@rit.edu    drsandeshghimire@gmail.com    rxlics@rit.edu

[1]Rochester Institute of Technology

## Abstract

Although iterative denoising (i.e., diffusion/flow) methods offer strong generative performance, they suffer from low generation efficiency, requiring hundreds of steps of network forward passes to simulate a single sample. Mitigating this requires taking larger step-sizes during simulation, thereby allowing one- or few-step generation. Recently proposed shortcut model learns larger step-sizes by enforcing alignment between its direction and the path defined by a base many-step flow-matching model through a self-consistency loss. However, its generation quality is significantly lower than the base model. In this paper, we formulate few-step generation as a controlled base generative process, and show that self-consistency loss can be understood through the lens of optimal control. This perspective naturally motivates its generalization to the proposed cumulative self-consistency loss that cumulatively penalizes misalignment along the entire trajectory. This encourages larger step-sizes that not only align with the base model at the current time step but also support alignment in the subsequent steps, facilitating high-quality generation. Furthermore, we draw a connection between our approach and reinforcement learning, potentially opening the door to a new set of approaches for few-step generation. Experiments show that we significantly improve one- and few-step generation quality under the same training budget. Implementation is available at: https://github.com/paribeshregmi/Shortcut-CSL

## 1 Introduction

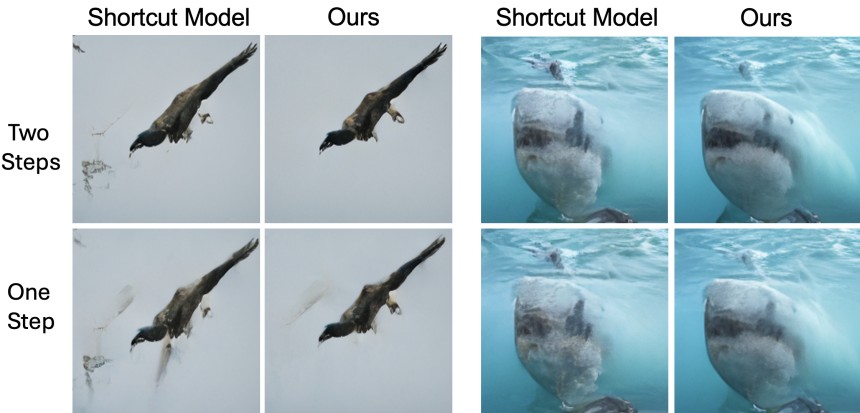

Figure 1: Two- and one-step image generation using Shortcut model (Frans et al., 2025) and our Shortcut-CSL method trained on ImageNet-256 dataset. Shortcut model tends to develop artifacts in the generated image with one and two-step generation, and ours generates a smooth, clearer image.

Diffusion (Song & Ermon, 2019; Ho et al., 2020; Song et al., 2021) and flow-matching (Lipman et al., 2023; Albergo et al., 2023) models have demonstrated remarkable capabilities in generating

---

*Corresponding Author

high-quality images and video (Rombach et al., 2022; Luo et al., 2023a;b), audio (Huang et al., 2023; Liu et al., 2023a), and molecular graphs (Jo et al., 2022; Eijkelboom et al., 2024). The generation involves iteratively transforming random noise into structured data, typically requiring hundreds of steps of forward passes through a neural network. This renders the process inefficient and computationally expensive. It is a key limitation in comparison to single-step generative models, such as VAE (Kingma & Welling, 2014; Vahdat & Kautz, 2020; Regmi & Li, 2023), GAN (Goodfellow et al., 2014), and normalizing flows (Dinh et al., 2017).

To achieve one-step generation, previous efforts train a student model to learn a direct mapping from noise-image pairs generated by the pre-trained model (Luhman & Luhman, 2021; Zhao et al., 2023; Zheng et al., 2023; Yin et al., 2024). A set of approaches avoid the cost of generating such pairs, and support flexible generation budgets by allowing few-step generation (Berthelot et al., 2023; Ghimire et al., 2023; Liu et al., 2023b; 2024; Salimans & Ho, 2022; Song et al., 2023). These methods either progressively distill knowledge by halving the number of generation steps at each stage (Salimans & Ho, 2022), or enforce straighter flow paths during distillation (Liu et al., 2023b; Lee et al., 2024). These distillation-based approaches require two training phases: training a base diffusion model and distilling its knowledge into a student model. Approaches like Consistency models (Song et al., 2023; Song & Dhariwal, 2024) and Meanflow (Geng et al., 2025) propose a single-phase training for few-step generation by learning both the base model and the few-step model in a single training pass. Recently, shortcut models, a family of models with step-sizes larger than that of the base models, show promising performance by conditioning the model's output not only on time-steps and noisy inputs but also on the simulation step-sizes (Frans et al., 2025). This simple but elegant design allows for generation under a specified budget simply by conditioning the model on its corresponding step size. They jointly train base model and the shortcut models with a standard flow-matching loss and a self-consistency loss (SL), which aligns the shortcut model's outputs with the base model at each time step. Although the experiments show improved few-step generation performance, it still lags significantly behind the base flow-matching models, emphasizing the need for further improvement. Moreover, the introduction of the SL loss in the paper lacks a theoretical foundation.

In this paper, we formulate few-step generation as a controlled base generative process and show that the SL loss admits a principled interpretation through the lens of the optimal control framework (Bryson & Ho, 1975). Through this viewpoint, we identify a critical limitation in the SL loss, and introduce a generalized version. Specifically, we show that the SL loss corresponds to a special case of the optimal control objective, which penalizes misalignment between the shortcut and the base model only at the current time step, overlooking the downstream effects of this immediate deviation. By drawing on the notion of expected future cost in a controlled process, we propose a general cumulative self-consistency loss (CSL) that penalizes misalignment across the entire trajectory, enabling the few-step model to account for these downstream effects. Moreover, we establish a connection between our algorithm and on-policy reinforcement learning methods (Mnih et al., 2013; Lillicrap et al., 2015) by interpreting the expected future cost as a value function. Similar to on-policy algorithms aiming to maximize an agent's cumulative reward rather than optimizing for immediate gains, our approach optimizes the cumulative cost along the generation trajectory. Evaluating the generative performance on standard benchmarks shows that our method robustly outperforms the state-of-the-art single-phase training approaches with the same training budget. Our contributions can be summarized as follows:

- We propose a novel perspective that frames few-step generation as a controlled process, and interpret the SL loss as a special case of an optimal control objective.
- We propose a novel CSL loss that penalizes both immediate and future misalignments between the few-step model and the base generative model cumulatively over time.
- Training with the proposed CSL loss significantly outperforms shortcut models and other few-step generation baselines on benchmark datasets.

## 2 RELATED WORKS

**Few-Step Generation:** Distillation-based approaches aim to transfer knowledge from a pretrained model to a student model (Luhman & Luhman, 2021; Zhao et al., 2023; Zheng et al., 2023; Yin et al., 2024; Liu et al., 2023b; Salimans & Ho, 2022). Some methods allow only fixed-step generation or require generating large volumes of noise-image pairs (Luhman & Luhman, 2021; Zhao et al., 2023;

Zheng et al., 2023; Yin et al., 2024; Liu et al., 2023b), while others reduce steps via bootstrapping but demand retraining a new model for each step reduction (Salimans & Ho, 2022). Overall, these approaches are either computationally expensive or inefficient when scaling to different step counts. Moreover, all these approaches follow a two-phase pipeline: pretraining a base model followed by distilling knowledge into few-step models. In contrast, consistency models (Song et al., 2023; Song & Dhariwal, 2024; Lu & Song, 2025) introduce a single-phase training strategy that trains a model to map noisy inputs from any time step directly to the final denoised version, essentially enforcing transformation consistency along the denoising trajectory. Remarkably, this class of models achieve one- or few-step performance close to the base diffusion or flow-matching model. However, they are complicated by design and naturally unstable while training, requiring substantial engineering efforts like adaptive weighing for training stability Lu & Song (2025). Recently, Frans et al. (2025) proposed shortcut models for few-step generation, featuring a simple, intuitive design and a straightforward single-phase training recipe, making them accessible for real-world deployment. Shortcut models condition not only on the noisy input and time step but also on the step size. By learning a family of denoising functions conditioned on the step size, they provide a flexible and efficient framework for few-step generation using a single model across different computation budgets. The SL loss is employed to align the generation trajectory of the model at larger step sizes with that of the base model, yielding strong performance. Nevertheless, the performance of few-step models still lags considerably behind that of the base model.

We establish a principled theoretical foundation for few-step generation from a control-theoretic viewpoint, casting the self-consistency (SL) loss as a special case of an optimal control objective. Whereas prior work formulates training objectives that supervise the few-step model at a single point in its trajectory – treating each step in isolation from its downstream consequences, this viewpoint leads us to a trajectory-level cumulative objective that explicitly accounts for these downstream effects. In addition, our theoretical framework uncovers a previously unidentified connection between few-step generation and reinforcement learning.

**Leveraging Control Theory to Guide Diffusion-Based Generation:** A few works have adopted control-theoretic principles to steer or fine-tune base diffusion and flow-matching models (Wang et al., 2025; Sprague et al., 2024; Domingo-Enrich et al., 2025). OC-Flow (Wang et al., 2025) leverages an optimal control framework to guide pre-trained flow-matching models for controlled generation. Domingo-Enrich et al. (2025) introduce a fine-tuning method that interprets reward-based optimization through the lens of stochastic optimal control.

## 3 BACKGROUND

### 3.1 OPTIMAL CONTROL

Let $X_t^u$ denote the state of a controlled dynamical system at time $t$ defined by:

$$dX_t^u = \big(b(X_t^u, t) + \sigma(t)\, u_\theta(X_t^u, t)\big)\, dt, \quad X_0^u \sim p_0, \tag{1}$$

where $b(X_t^u, t)$ is the base drift, $u_\theta(X_t^u, t)$ is called the control vector field, $\sigma(t)$ is a control coefficient, and $p_0$ is the initial distribution. These jointly define the evolution of the controlled process $X_t^u$. Optimal control problem aims to minimize the following objective, which is also called a *value function*:

$$J(u_\theta; x_s, s) = \int_s^1 \Big( f\big(u_\theta(X_t^u, t), t\big) + g(X_t^u, t) \Big)\, dt + h(X_1^u), \quad x_s = X_s^u \tag{2}$$

where $f\big(u_\theta(X_t^u, t), t\big)$ penalizes the magnitude of the control, $g(X_t^u, t)$ is the intermediate state cost, and $h(X_1^u)$ is the terminal cost. A result which we will later use in our analysis, is the expression for the derivative of $J(u_\theta; x_s, s)$ with respect to the control parameters $\theta$ is:

$$\frac{dJ(u_\theta; x_s, s)}{d\theta} = \int_s^1 \frac{\partial}{\partial\theta} f\big(u_\theta(X_t^u, t), t\big)\, dt + \int_s^1 \frac{\partial u_\theta(X_t^u, t)}{\partial\theta} \sigma(t) \nabla_{x_t} J(u_\theta; x_t, t)\, dt \tag{3}$$

The gradient in Equation (3) consists of an immediate and a propagated effect. The first term measures how changes in control parameter $\theta$ directly influence the instantaneous control cost, while the second term captures how those same changes alter the future trajectory and thus the future costs. Together, they quantify both the local and downstream impacts of the change in $\theta$.

## 3.2 FLOW MATCHING

Flow-matching models (Lipman et al., 2023; Albergo et al., 2023) formulate data generation as simulating an ordinary differential equation (ODE), iteratively converting random noise into realistic data samples. Given a model with parameter $\theta$ that outputs velocity field $v_\theta$, the denoising process is represented as:

$$dx_t = v_\theta(x_t, t)\, dt\,, \quad t \in [0, 1],\; x_0 \sim \mathcal{N}(0, I)$$

The goal is to train the model such that the distribution of samples at $t = 1$ matches the data distribution. We consider the optimal transport formulation of flow-matching (Lipman et al., 2023) in which the model is trained to optimize the following objective:

$$\mathcal{L}_{FM}(\theta) = \mathbb{E}_{x_0, x_1} \left[ \|v_\theta(x_t, t) - v_t\|^2 \right]; \quad x_t = (1 - t)x_0 + tx_1, \quad v_t = x_1 - x_0 \tag{4}$$

Here, $x_1$ is a sample from the data distribution, $x_1 \sim D$, and $v_t$ represents the direction from noise to the data sample. Multiple $(x_0, x_1)$ pairs may correspond to the same $x_t$, making $v_t$ inherently stochastic. When trained with the objective in Equation (4), flow-matching models learn the expected velocity conditioned on $x_t$, denoted as $\mathbb{E}[v_t | x_t]$.

## 3.3 SHORTCUT MODELS

The trajectory defined by the flow-matching ODE is typically curved, necessitating small step sizes for accurate simulation, resulting in inefficient generation. Using larger step sizes to enable faster, few-step generation leads to deviations from the true trajectory and significantly degrades sample quality, often causing failure when generating with fewer than four steps. To overcome this limitation, Frans et al. (2025) propose shortcut models, which learn to take larger steps while accounting for the future curvature of the trajectory, thereby avoiding deviation from the original path. This approach enables few-step generation with improved sample quality. Notably, both the flow-matching and shortcut models are parameterized by the same backbone network, with different variants conditioned on the step size. A step taken by the shortcut model is given by:

$$x_{t+d} = x_t + \mathcal{S}(x_t, t, d) \cdot d \tag{5}$$

where $d$ is the step size, and $\mathcal{S}(x_t, t, d)$ denotes the normalized direction from $x_t$ to the point $x_{t+d}$ on the trajectory. Shortcut models of different step sizes $d$ can be trained on the same backbone network. The flow-matching model is the one conditioned with $d = 0$ i.e., $\mathcal{S}(x_t, t, 0)$. Shortcut models are trained to enforce consistency across step sizes: taking a single step of size $2d$ should be equivalent to taking two sequential steps of size $d$.

$$\mathcal{S}(x_t, t, 2d) = \mathcal{S}_{\text{target}}, \quad \mathcal{S}_{\text{target}} = \frac{\mathcal{S}(x_t, t, d)}{2} + \frac{\mathcal{S}(x_{t+d}, t + d, d)}{2} \tag{6}$$

The target for the shortcut model $\mathcal{S}(x_t, t, 2d)$ is thus generated by bootstrapping. The joint training objective of the flow-matching and shortcut models is:

$$\mathcal{L}_{\text{shortcut}}(\theta) = \mathbb{E}_{x_0, x_1, t, d} \Big[ \underbrace{\|\mathcal{S}_\theta(x_t, t, 0) - (x_1 - x_0)\|^2}_{\text{Flow-Matching}} + \underbrace{\|\mathcal{S}_\theta(x_t, t, 2d) - \mathcal{S}_{\text{target}}\|^2}_{\text{Self-Consistency}} \Big] \tag{7}$$

## 4 CONTROL THEORETIC FORMULATION OF FEW-STEP MODELS

Both the forward and backward processes in flow-matching and diffusion models are governed by the Fokker–Planck equation, which describes the evolution of a probability density over time (Risken, 1996). From this perspective, one- or few-step generation can be viewed as training a neural network to approximate the solution of an underlying ODE. Consequently, when training a shortcut model to approximate this solution, the resulting integration error is cumulative in nature. To formalize this intuition, we model the output of the shortcut model itself as the solution to an ODE, defined as follows:

$$dX_t^u = \left[ b(X_t^u, t) + u_\theta(X_t^u, t) \right] dt \tag{8}$$

This formulation implies that the shortcut model's output incurs an error $u_\theta(X_t^u, t)$ at each point in time that makes its trajectory deviate away from the base model. Consequently, the error at time $t$ reflects the cumulative effect of all errors at earlier time points $\tau < t$. Note that the model predicts the shortcut velocity $dX_t^u/dt$, and the error $u_\theta$ is obtained by comparing this prediction with the base velocity $b(X_t^u, t)$. Another advantage is that Equation (8) provides a new perspective of viewing the few-step model as a controlled process. Unlike the standard optimal control formulation, where a control vector field $u_\theta$ is introduced to steer the base process toward optimizing a predefined objective (e.g., Equation (2)), here $u_\theta$ is an unknown error that is implicitly introduced along the generation trajectory of a shortcut model. Since our goal is solely to make the shortcut model's trajectory match the base model, we focus on minimizing the error $u_\theta$ and do not consider intermediate costs $g(X_t^u, t)$ or terminal cost $h(X_1^u)$. Accordingly, we define an objective by setting $g = h = 0$ in Equation (2).

$$J(u_\theta; x_s, s) = \int_s^1 f\big(u_\theta(X_t^u, t), t\big)\, dt \tag{9}$$

The future cost, $J$, thus becomes the primary quantity of interest. It is important to note that if $u_\theta = 0$ for all $t$ and $X$, the controlled process matches exactly with the base process, and the path of the few-step model matches that of the base flow-matching model.

Equations (8) and (9) reveal that changing the parameters $\theta$ affects cumulative objective $J$ in two ways: (i) by altering the immediate error $u_\theta(X_s^u, s)$, and (ii) by changing the subsequent states $X_t^u$, which in turn impacts the future errors $u_\theta(X_t^u, t)$ for $t > s$. In the next section, we show that SL loss is a special case that accounts only for the immediate error, and propose its two generalizations.

## 4.1 SELF-CONSISTENCY LOSS (SL) AS AN OPTIMAL CONTROL OBJECTIVE

**Lemma 1** Let $f\big(u_\theta(X_t^u, t), t\big) = \|u_\theta(X_t^u, t)\|^2 \boldsymbol{\delta}(s-t)$, where $\boldsymbol{\delta}(t)$ is the Dirac delta function. The objective in Equation (9) becomes:

$$J_{SL}(u_\theta; x_s, s) = \int_s^1 \|u_\theta(X_t^u, t)\|^2 \boldsymbol{\delta}(s-t)dt = \|u_\theta(x_s, s)\|^2, \quad x_s = X_s^u \tag{10}$$

From Equation (10), if we consider model with stepsize $2d$ as shortcut model $\frac{dX_t^u}{dt}$ and that with stepsize $d$ as base drift $b(X_t^u, t)$, then for the discrete case, $u_\theta$ is the difference between the resultant of the two steps taken by the base model and a single step taken by the shortcut model: $u_\theta(x_t, t) = \mathcal{S}_\theta(x_t, t, 2d) - \mathcal{S}_{\text{target}}$. The optimal control objective equals to the self-consistency loss term:

$$J_{SL}(u_\theta; x_t, t) = \|\mathcal{S}_\theta(x_t, t, 2d) - \mathcal{S}_{\text{target}}\|^2$$

Equation (9) accumulates the cost $f$ over the full trajectory from $t = s$ to $t = 1$, but SL restricts this to only the current time $t = s$ via a Dirac delta function. Thus, $J_{SL}$ is suboptimal: it penalizes error $u_\theta$ only at $t = s$, ignoring how it propagates and affects future errors at $t > s$ along the trajectory.

## 5 CUMULATIVE SELF-CONSISTENCY LOSS (CSL)

We relax the delta function constraint in $J_{SL}$ and propose the following two losses:

**Lemma 2** Let, $f\big(u_\theta(X_t^u, t), t\big) = \|u_\theta(x_s, s)\|^2 \ \forall t > s$, where, $x_s = X_s^u$ the objective in Equation (9) becomes:

$$J_{USL}(u_\theta; x_s, s) = \int_s^1 \|u_\theta(X_t^u, t)\|^2 dt = (1-s)\|u_\theta(x_s, s)\|^2 \tag{11}$$

where USL stands for Uniform Self-Consistency Loss. $J_{USL}$ considers errors over the entire time $s < t < 1$; however, it makes a naive assumption that the error $u_\theta(X_t^u, t)$ is independent of $t$, and is uniform over the entire trajectory, which is equal to the value at $t = s$. Essentially, this simply reduces to a weighted SL and suffers from the same limitation as $J_{SL}$. We further relax the uniformity assumption and propose a CSL loss.

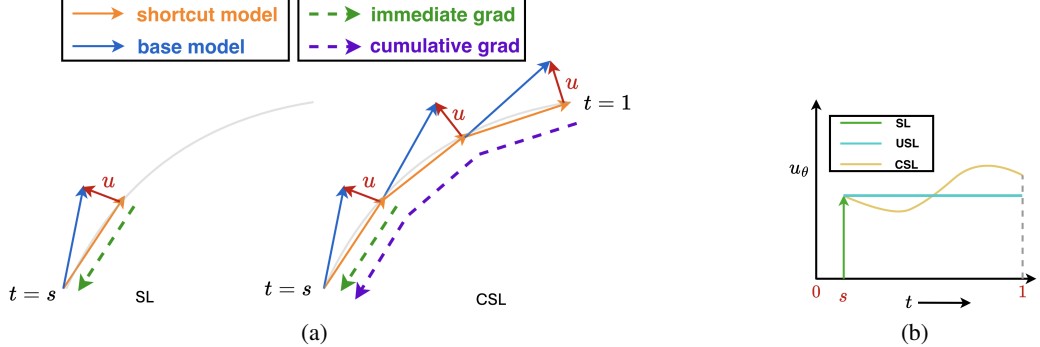

Figure 2: (a)Illustration of the SL (left) and CSL (right) losses. The solid red arrow denotes error $u_\theta$, which is the difference between the direction taken by the base model and the shortcut model. For SL, we calculate the error only at the current time $s$, while for CSL, we calculate and aggregate the error at every point from $t = s$ to the final time $t = 1$. CSL allows the model to learn from cumulative gradient in addition to the immediate gradient as denoted by dashed purple and green arrows respectively. (b) A comparison between SL, USL, and CSL losses.

**Lemma 3** *Let $f\big(u_\theta(X^u_t, t), t\big) = \|u_\theta(X^u_t, t)\|^2, \forall t > s$, the objective in Equation (9) becomes:*

$$J_{CSL}(u_\theta; x_s, s) = \int_s^1 \|u_\theta(X^u_t, t)\|^2 dt, \quad x_s = X^u_s \tag{12}$$

The differences between the SL, USL, and CSL losses are illustrated in Figure 2(b). Starting from $t = s$, SL penalizes the error $u_\theta$ only at that specific time. USL accounts for the error along the entire trajectory; however, it assumes the error is uniform across time, effectively treating the error at all points as identical to that at $t = s$. In contrast, CSL captures the true error landscape along the entire trajectory, accumulating the error at each time step.

## 5.1 ANALYSIS AND COMPARISON WITH SELF-CONSISTENCY LOSS

From Equation (3), the gradient of $J_{CSL}$ with respect to parameters $\theta$ is given as (for $\sigma(t) = 1$):

$$\frac{dJ_{CSL}(u_\theta; x_s, s)}{d\theta} = \int_s^1 \left(2u_\theta + \nabla_{x_t} J_{CSL}(u_\theta; x_t, t)\right) \frac{\partial u_\theta}{\partial \theta} dt$$

Analyzing the integrand,

$$\left(2u_\theta + \nabla_{x_t} J_{CSL}(u_\theta; x_t, t)\right)\frac{\partial u_\theta}{\partial \theta} = \left(\underbrace{2u_\theta}_{\text{immediate grad}} + \underbrace{\nabla_{x_t} \int_t^1 \|u_\theta(X^u_t, t)\|^2 dt}_{\text{cumulative grad}}\right)\frac{\partial u_\theta}{\partial \theta} \tag{13}$$

In Equation (13), the gradient of $J_{CSL}$ comprises two components: an immediate gradient term corresponding to the error at the current step, and a cumulative gradient term corresponding to the cumulative error along the trajectory. In contrast, the gradient of the $J_{SL}$ and $J_{USL}$ include only the immediate gradient. Note that the cumulative gradient is a gradient of the future errors with respect to the state $x_t$. Optimizing $J_{CSL}$ therefore encourages the shortcut model not only to align with the base model at the current step, but also to guide the trajectory towards a state $x_t$ that facilitates high-quality generation by supporting the alignment in the subsequent steps as well.

## 5.2 CSL ESTIMATION AND TRAINING

Since we are working with discrete time steps, the loss $J_{CSL}$ in Equation (12) can be expressed as a summation of the norm of $u_\theta$ at every discrete step along the trajectory:

$$J_{CSL}(u_\theta; x_{nd}, nd) = \sum_{k=n}^{R'} \|u_\theta(x_{dk}, dk, d)\|^2, \quad R' = \frac{1}{d}, \quad n \in \{1, 2, \ldots, R' - 1\} \tag{14}$$

Let $R = R' - n + 1$ denote the number of terms in the summation. Replacing $R' = \frac{1}{d}$ with $R' = n$ (i.e. $R = 1$) results in an SL objective. An illustration of the SL and CSL losses is presented in Figure 2. Estimating summation in Equation (14) requires simulation, which can be computationally demanding. However, we can approximate the summation by including only a few terms. Specifically, we find that using only two terms, i.e., $R = 2$, significantly improves few-step generation performance with negligible computational overhead. The effectiveness of $R = 2$ estimation is because for few-step models with two or four steps, even a two-step simulation covers a substantial portion of the trajectory, covering the full trajectory for a 2-step model and 50% of the trajectory for a 4-step shortcut model. We have presented the Algorithm, and details of how gradient backpropagates from future ($k > n$) steps to the current ($k = 1$) step in Appendix Section E.

### 5.3 CONNECTION TO REINFORCEMENT LEARNING

In our setting, the reinforcement learning analogy can be drawn by viewing the agent as attempting to transform noise into a data sample, where the states correspond to intermediate noisy samples along the generation path, the action is the direction of the next step, and the reward is defined as the negative magnitude of $u_\theta$. Under this view, the objective $J_{CSL}$ in Equation (12) plays a role similar to a value function: it quantifies the cumulative cost of following a policy along the trajectory. Moreover, since few-step generation involves a small, fixed number of steps, we do not require a separate network to approximate $J_{CSL}$; instead, it can be estimated directly by rolling out the actions. A more detailed discussion on this connection is provided in the Appendix Section F.

Table 1: Comparison of FID-50K ↓ scores of the baselines and our method on CelebA-256 and CIFAR10. (* indicates that results are taken from Frans et al. (2025)

|  | CelebA-256 | | | | CIFAR10 | | | |
|---|---|---|---|---|---|---|---|---|
|  | 128-Step | Four-Step | Two-Step | One-Step | 128-Step | Four-Step | Two-Step | One-Step |
| **Two Phase Training** | | | | | | | | |
| **Reflow** | 12.80±0.03 | 13.77±0.05 | 14.48±0.05 | 16.07±0.02 | 13.93±0.05 | 14.92±0.05 | 15.59±0.14 | 16.98±0.06 |
| **PD** | 7.96±0.07 | 14.49±0.07 | 16.73±0.02 | 20.40±0.15 | 7.89±0.07 | 10.75±0.09 | 11.80±0.08 | **13.26±0.09** |
| **Single Phase Training** | | | | | | | | |
| **FM** | 7.92±0.04 | 62.8±0.04 | 112.1±0.07 | 321.2±0.04 | 7.95±0.09 | 65.03±0.03 | 177.9±0.12 | 385.1±0.11 |
| **CT\*** | 53.7 | 19.0 | - | 33.2 | - | - | - | - |
| **ST** | **7.83±0.04** | 9.36±0.05 | 12.56±0.02 | 20.46±0.02 | 7.37±0.03 | 9.15±0.13 | 11.79±0.07 | 19.80±0.03 |
| **ST-USL** | 7.95±0.02 | 9.18±0.08 | 12.00±0.05 | 19.41±0.03 | 7.37±0.08 | 9.35±0.07 | 11.65±0.08 | 19.57±0.05 |
| **ST-CSL** | 7.88±0.04 | **8.98±0.02** | **10.96±0.02** | **18.37±0.02** | **7.13±0.03** | **8.10±0.09** | **9.24±0.09** | 17.76±0.02 |

## 6 EXPERIMENTS

In this section, we empirically evaluate the effect of proposed CSL on performance. In Section 6.1 we compare the generative performance of our method with the baselines on the unconditional generation task on CelebA256 (Liu et al., 2015) and CIFAR10 (Krizhevsky, 2009). In Sections 6.2 and 6.3, we assess if the improvement brought by CSL is consistent across varying backbone network size and varying ratio of flow-matching to bootstrap targets ($B : K$) (see Algorithm 1) along with the training time comparison of the methods. In Section 6.4 we assess the effect of increasing the number of terms $R$ on the performance. Finally, in Section 6.5 we evaluate the baselines and or model on class-conditional generation in ImageNet256 benchmark. For evaluations, we use FID-50K and F1 scores as the comparison metrics (metric details are provided in the Appendix Section H).

### 6.1 PERFORMANCE COMPARISON

We consider two categories of baselines for performance comparison. **1. Two-Phase distillation approaches:** Progressive Distillation (PD) (Salimans & Ho, 2022), Reflow (Liu et al., 2023b).

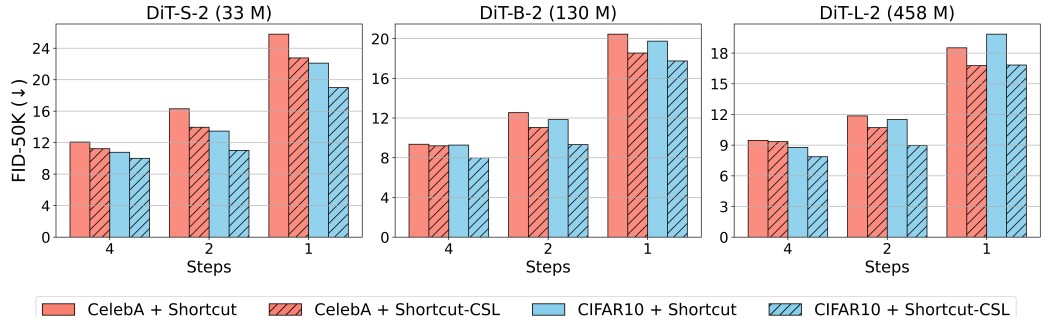

Figure 3: FID score comparison between shortcut and shortcut-CSL (ours) across different backbone network sizes (parameter counts). The networks are diffusion transformers (DiT) of varying sizes. Our model consistently outperforms the shortcut model across all network sizes and generation steps on both CelebA-256 and CIFAR10 datasets.

**2. Single-Phase training approaches:** Flow-Matching (FM) (Lipman et al., 2023), Consistency Training (CT) (Song et al., 2023), Shortcut Models (ST)(Frans et al., 2025)

For a fair comparison with the shortcut model, we make sure the training budget is the same for both methods. We use the same number of flow-matching and bootstrap targets per batch. To achieve this, since we create two bootstrap targets per datapoint for $R = 2$, we only make use of $K/2$ datapoints for bootstrap targets in our case, while for the shortcut model we use $K$ datapoints. We require additional compute for the additional network forward pass step during bootstrap targets generation (step 18 in Algorithm 1), which adds only $6 - 10\%$ of extra computation and training time in practice. Similar to Frans et al. (2025), we use the medium-scale diffusion transformer model DiT-B-2 (Peebles & Xie, 2023) as a backbone network. All the other hyperparameters are the same, and the details are provided in the Appendix. The FID-50K scores for the baselines and our method are reported in Table 1, the base flow-matching model is the one with 128-steps. Our method, trained with $J_{USL}$ and $J_{CSL}$, are referred to as shortcut-USL (ST-USL) and shortcut-CSL (ST-CSL), respectively. The results show that ST-USL consistently outperforms the baseline ST in one-, two-, and four-step generation, except for the four-step case on CIFAR10. Notably, the performance of ST-CSL surpasses both ST and ST-USL on both datasets across all the steps, with the exception of the base 128-step model on CelebA, where it achieves performance comparable to ST. This shows the effectiveness of using CSL in improving the generation quality of few-step models. Additionally, while PD achieves superior one-step generation performance on CIFAR-10, and Reflow does so on CelebA, both methods require a two-phase training procedure. Furthermore, they offer a less favorable trade-off between computational cost and performance, underperforming on 2- and 4-step generation tasks compared to our method. PD also requires training and deploying separate models for each generation budget, which is time-inefficient.

Next, we empirically asses the analysis in Section 5.1 that our method trains the model to not only minimize current but also future errors. We compare trained Shortcut and Shortcut-CSL models for CIFAR10 by generating samples from random noise at $t = 0$ and measuring the misalignment $u_\theta$ between the two-step and it's base four-step trajectories at intermediate time steps $t = 0.5$ and $t = 1.0$. For a

Table 2: Squared misalignment $(u_\theta^2)$ for CIFAR-10, averaged over 100 samples.

| Method | $t = 0.5$ | $t = 1.0$ |
|---|---|---|
| **ST** | $0.5 \times 10^{-3}$ | $2.5 \times 10^{-3}$ |
| **ST-CSL** | $0.5 \times 10^{-3}$ | $1.4 \times 10^{-3}$ |

two-step trajectory, the value of $u_\theta$ at $t = 0.5$ and $t = 1.0$ corresponds to immediate and future errors respectively. Table 2 shows that while both methods have similar value of $u_\theta$ at $t = 0.5$, Shortcut-CSL has a lower value at $t = 1.0$, demonstrating its advantage in reducing overall error.

## 6.2 PERFORMANCE WITH INCREASING BACKBONE NETWORK SIZE

We evaluate whether the improvement of ST-CSL over ST is consistent across different backbone network sizes. We train both models using small (DiT-S-2, 33M parameters), medium (DiT-B-2, 130M parameters), and large (DiT-L-2, 458M parameters) diffusion transformer networks.

The FID scores for models trained on CelebA and CIFAR10 are shown in Figure 3. We observe that as the model size increases, the overall few-step generation quality of the shortcut model improves.

Importantly, ST-CSL consistently outperforms ST across all model sizes and generation steps. This shows that the improvement of CSL over SL is robust across varying network sizes.

**Training Time Comparison:** In Table 3, we compare the time taken to train the flow-matching (FM), shortcut(ST), and our methods. Training FM is efficient of all but it doesn't support few-step generation. Compared to ST, our method only consumes $6 - 10\%$ more training time on average. Please refer to the Appendix Section G for detailed time and memory consumption analysis.

Table 3: Wall-clock time (in hours) for training (100 epochs) an FM, ST, and ST-CSL methods with different sizes Diffusion Transformers on the CelebA dataset.

|  | FM | ST | ST-CSL |
|---|---|---|---|
| DiT-S-2 (33M) | – | 5.6 | 6.2 |
| DiT-B-2 (130M) | 6.4 | 7.4 | 7.8 |
| DiT-L-2 (458M) | – | 11.4 | 12.2 |

## 6.3 ANALYSING THE EFFECT OF THE RATIO $B : K$

We further evaluate whether the improvement of ST-CSL is consistent if we vary the ratio of flow-matching to bootstrap targets ($B : K$) while training. We report the FID scores with varying ratio in Table 4. Although increasing the percentage of bootstrap targets introduces additional computational cost, our results show that it significantly enhances the generation performance. Importantly, our ST-CSL consistently outperforms the ST baseline in few-step generation across all ratios. Moreover, we observe that at the ratio $1 : 1$, our method achieves two-step performance within 2.0 points of ST's 128-step performance on CelebA, and within 1.5 points on CIFAR-10.

Table 4: FID score comparison between shortcut and shortcut-CSL (ours) for different ratios of flow-matching to bootstrap targets ($B : K$). Our model consistently outperforms shortcut model across all ratios and generation steps on both CelebA-256 and CIFAR10 datasets.

| B:K | Method | CelebA-256 | | | | CIFAR10 | | | |
|---|---|---|---|---|---|---|---|---|---|
|  |  | 128-Step | Four-Step | Two-Step | One-Step | 128-Step | Four-Step | Two-Step | One-Step |
| 4:1 | **ST** | **7.83** | 9.36 | 12.56 | 20.46 | 7.37 | 9.15 | 11.79 | 19.80 |
|  | **ST-CSL** | 7.88 | **8.98** | **10.96** | **18.37** | **7.13** | **8.10** | **9.24** | **17.76** |
| 2:1 | **ST** | **7.73** | 9.00 | 11.51 | 18.50 | 6.98 | 8.53 | 11.00 | 18.33 |
|  | **ST-CSL** | 7.75 | **8.58** | **10.09** | **16.37** | **6.95** | **7.98** | **9.08** | **16.51** |
| 1:1 | **ST** | 8.01 | 9.00 | 11.14 | 16.51 | **6.56** | 8.17 | 10.54 | 16.43 |
|  | **ST-CSL** | **7.56** | **8.54** | **9.91** | **15.51** | 6.67 | **6.95** | **7.94** | **13.96** |

## 6.4 EFFECT OF $R$

We investigate the effect of increasing the number of terms in the CSL objective in Equation (14). Specifically, we experiment with $R = 1, 2, 4$, where larger values of $R$ yield more accurate estimations of the CSL. Note that $R = 1$ corresponds to the SL loss used in the shortcut models. For all settings of $R$, the number of flow-matching and bootstrap targets per batch is kept constant. We observe that, compared to $R = 1$, $R = 2$ incurs approximately a $5\%$ more training time, while $R = 4$ incurs about $30\%$ more time. The FID scores for few-step generation under differ-

Table 5: FID-50K scores for one- and few-step generation for different values of $R$ on CI-FAR10.

| | CIFAR10 | | |
|---|---|---|---|
| $R$ | Four-Step | Two-Step | One-Step |
| 1 | $8.17 \pm 0.10$ | $10.54 \pm 0.08$ | $16.43 \pm 0.06$ |
| 2 | $6.95 \pm 0.07$ | $7.94 \pm 0.07$ | $13.96 \pm 0.10$ |
| 4 | $\mathbf{6.66} \pm \mathbf{0.11}$ | $\mathbf{7.11} \pm \mathbf{0.05}$ | $\mathbf{13.10} \pm \mathbf{0.06}$ |

ent values of $R$ are reported in Table 5. The results demonstrate that increasing $R$ consistently improves generation performance, highlighting the benefit of more accurate CSL estimation.

## 6.5 CLASS CONDITIONAL GENERATION ON IMAGENET

We would like to evaluate further the efficacy of the proposed CSL on the class-conditional generation task. For this, we train the baseline ST and our ST-CSL methods on the ImageNet dataset, and calculate the FID and the F1 scores of the generated samples. For this analysis, we also compare with the recent MeanFlow (Geng et al., 2025) baseline at different percentages of bootstrap target, which corresponds to the $B : K$ ratio for ST and ST-CSL. The FID for base 128-step generation is 15.21. The results in Table 6 show significant gains of our ST-CSL method when compared to Meanflow and ST. Additionally, although Meanflow performs better than ST in one-step generation, ST offers better performance vs efficiency tradeoff as shown by its superior performance at two- and four-step generations. Moreover, compared to Table 1, the gain of ST-CSL over ST is even larger in advanced and challenging datasets like ImageNet-256: for one-step generation, our method improves FID by nearly 20 points and F1 score by 4%.

Furthermore, in Figure 1, images generated by ST with one and two steps have noticeable artifacts; for instance, in the flying-bird example, despite the plain background, distinct distortions appear around the object. In contrast, our (ST-CSL) method yields smoother and cleaner generations.

Table 6: FID and F1 score comparison between baselines our method on the class-conditional generation task on ImageNet-256, trained on DiT-B-2 backbone. We compare the performance across different ratios $B : K$ for ST and ST-CSL, and different percentage of bootstrap targets for Meanflow. ST-CSL consistently outperforms ST and Meanflow across all ratios and generation steps.

| Method | FID ↓ | | | F1-score ↑ | | |
|---|---|---|---|---|---|---|
| | Four-Step | Two-Step | One-Step | Four-Step | Two-Step | One-Step |
| **Reflow** | 40.86 | 43.68 | 50.40 | 0.56 | 0.54 | 0.51 |
| **FM** | 104.96 | 210.39 | 325.78 | 0.26 | 0.09 | 0.00 |
| **Meanflow (15%)** | 48.22 | 50.13 | 59.63 | 0.55 | 0.55 | 0.54 |
| **Meanflow (10%)** | 39.65 | 41.10 | 50.01 | 0.57 | 0.57 | 0.55 |
| **Meanflow (5%)** | 34.09 | 36.61 | 45.12 | 0.58 | 0.57 | **0.56** |
| **ST (4:1)** | 24.70 | 35.73 | 64.12 | 0.61 | 0.56 | 0.46 |
| **ST (2:1)** | 23.47 | 32.54 | 55.55 | 0.63 | 0.58 | 0.50 |
| **ST (1:1)** | 24.17 | 32.22 | 51.78 | 0.63 | 0.59 | 0.51 |
| **ST-CSL (4:1)** | 16.98 | 21.77 | 45.84 | 0.63 | 0.60 | 0.50 |
| **ST-CSL (2:1)** | 16.21 | 18.71 | 37.60 | 0.64 | 0.62 | 0.53 |
| **ST-CSL (1:1)** | **15.71** | **17.35** | **31.66** | **0.64** | **0.63** | **0.56** |

## 7 CONCLUSION AND FUTURE WORK

We formulate few-step generation as a controlled process, using the flow-matching model as its base process. This perspective provides the theoretical foundation for the self-consistency loss and motivates our proposed cumulative self-consistency loss. By training with this objective, we achieve significant improvement in one- and few-step generation performance over the baselines.

In this work, we consider the objective $J$ without any intermediate state cost, i.e., $g(X_t^u, t) = 0$. An interesting direction for future work is to explore ways of incorporating this cost to influence the trajectory by penalizing undesirable intermediate states. Furthermore, the connection of our formulation to reinforcement learning offers opportunities to leverage techniques from the reinforcement learning literature to further enhance few-step generation quality.

## 8 ACKNOWLEDGEMENT

This work is supported by the National Science Foundation under NSF Award No. 2045804 and the National Institute of General Medical Sciences of the National Institutes of Health under Award No. R35GM156653. We acknowledge Research Computing at the Rochester Institute of Technology (RC) for providing computational resources.

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

APPENDIX

## A   DERIVATION OF EQUATION (3)

We have,

$$
J(u_\theta; x_s, s) = \int_s^1 \Big( f\big(u_\theta(X_t^u, t), t\big) + g(X_t^u, t)\Big)\, dt + h(X_1^u), \quad x_s = X_s^u
$$

**Step 1: An expression for the gradient of $J$:**

$$
\frac{d}{d\theta} \int_s^1 f(u_\theta(X_t^u,t),t) + g(X_t^u,t)\, dt + h(X_1^u)
$$
$$
= \int_s^1 \frac{\partial}{\partial\theta} f(u_\theta(X_t^u,t),t) dt + \int_s^1 \frac{\partial[f(u_\theta(X_t^u,t),t) + g(X_t^u,t)]}{\partial X_t^u} \frac{\partial X_t^u}{\partial\theta}\, dt + \frac{\partial h(X_1^u)}{\partial\theta} \tag{15}
$$

**Step 2: Expression for gradients $\frac{dX_t^u}{dX_s^u}$ and then $\frac{dX_t^u}{d\theta}$**

The state evolves according to

$$
\frac{dX_t^u}{dt} = b(X_t^u, t) + \sigma(t)\, u_\theta(X_t^u, t)
$$

The state-to-state sensitivity is given by:

$$
A(t,s) := \frac{dX_t^u}{dX_s^u}, \quad t \geq s
$$

For a small time increment $\varepsilon > 0$ such that $s + \epsilon < t$, using the chain rule along the trajectory,

$$
\frac{dX_t^u}{dX_s^u} = \frac{dX_t^u}{dX_{s+\varepsilon}^u} \frac{dX_{s+\varepsilon}^u}{dX_s^u}
$$

In terms of $A(t,s)$,

$$
A(t,s) = A(t, s+\varepsilon)\, A(s+\varepsilon, s) \tag{16}
$$

Using a Taylor expansion for $X_{s+\varepsilon}^u$,

$$
X_{s+\varepsilon}^u = X_s^u + \varepsilon\big[b(X_s^u, s) + \sigma(s)u_\theta(X_s^u, s)\big] + O(\varepsilon^2)
$$

Thus,

$$
A(s+\varepsilon, s) = 1 + \varepsilon \frac{\partial}{\partial X_s^u}\big[b(X_s^u, s) + \sigma(s)u_\theta(X_s^u, s)\big] + O(\varepsilon^2) \tag{17}
$$

By definition of the derivative,

$$
\frac{\partial}{\partial s} A(t,s) = \lim_{\varepsilon\to 0^+} \frac{A(t, s+\varepsilon) - A(t,s)}{\varepsilon}
$$

Using Equations (16) and (17),

$$
\frac{\partial}{\partial s} A(t,s) = \lim_{\varepsilon\to 0^+} \frac{-\varepsilon A(t,s+\varepsilon)\frac{\partial}{\partial X_s^u}\big[b(X_s^u,s) + \sigma(s)u_\theta(X_s^u,s)\big] - O(\varepsilon^2)}{\varepsilon}
$$
$$
= -A(t,s)\frac{\partial}{\partial X_s^u}\big[b(X_s^u,s) + \sigma(s)u_\theta(X_s^u,s)\big]
$$

Therefore,

$$\frac{\partial}{\partial s}\frac{dX_t^u}{dX_s^u} = -\frac{dX_t^u}{dX_s^u}\frac{\partial}{\partial X_s^u}\big[b(X_s^u,s) + \sigma(s)u_\theta(X_s^u,s)\big]$$

$$\implies \frac{dX_t^u}{dX_s^u}\Big|_{s=s} = -\int_t^s \frac{dX_t^u}{dX_s^u}\frac{\partial}{\partial X_s^u}\big[b(X_s^u,s) + \sigma(s)u_\theta(X_s^u,s)\big]\,ds\ +\frac{dX_t^u}{dX_s^u}\Big|_{s=t} \quad (18)$$

Equation (18) can be generalized to obtain gradient with respect to the parameter $\theta$ (Chen et al., 2018), which is given by the following integral :

$$\frac{dX_t^u}{d\theta} = -\int_t^s \frac{dX_t^u}{dX_s^u}\frac{\partial}{\partial\theta}\big[b(X_s^u,s) + \sigma(s)\,u_\theta(X_s^u,s)\big]ds$$

$$= \int_s^t \frac{dX_t^u}{dX_s^u}\,\sigma(s)\,\frac{\partial u_\theta(X_s^u,s)}{\partial\theta}\,ds \quad (19)$$

## Step 3: Final Expression

Substituting the gradient representation from Equation (19) into Equation (15), we get:

$$\frac{dJ(u_\theta;x_s,s)}{d\theta} = \int_s^1 \frac{\partial}{\partial\theta}f\big(u_\theta(X_t^u,t),t\big)\,dt$$

$$+ \int_s^1 \frac{\partial[f(u_\theta(X_t^u,t),t) + g(X_t^u,t)]}{\partial X_t^u}\left(\int_s^t \frac{dX_t^u}{dX_{t'}^u}\,\sigma(t')\,\frac{\partial u_\theta(X_{t'}^u,t')}{\partial\theta}\,dt'\right)dt$$

$$+ \frac{\partial h(X_1^u)}{\partial X_1^u}\int_s^1 \frac{dX_1^u}{dX_{t'}^u}\,\sigma(t')\,\frac{\partial u_\theta(X_{t'}^u,t')}{\partial\theta}\,dt'$$

Interchanging the order of integration gives

$$\frac{dJ(u_\theta;x_s,s)}{d\theta} = \int_s^1 \frac{\partial}{\partial\theta}f\big(u_\theta(X_t^u,t),t\big)\,dt$$

$$+ \int_s^1 \left[\int_{t'}^1 \frac{\partial[f(u_\theta(X_t^u,t),t) + g(X_t^u,t)]}{\partial X_t^u}\frac{dX_t^u}{dX_{t'}^u}\,dt + \frac{\partial h(X_1^u)}{\partial X_1^u}\frac{dX_1^u}{dX_{t'}^u}\right]\sigma(t')\,\frac{\partial u_\theta(X_{t'}^u,t')}{\partial\theta}\,dt'$$

Recognizing the value gradient, $(x_s = X_s^u)$

$$\nabla_{x_{t'}}J(u_\theta;x_{t'},t') = \int_{t'}^1 \frac{\partial[f(u_\theta(X_t^u,t),t) + g(X_t^u,t)]}{\partial X_t^u}\frac{dX_t^u}{dX_{t'}^u}\,dt + \frac{\partial h(X_1^u)}{\partial X_1^u}\frac{dX_1^u}{dX_{t'}^u},$$

we arrive at the final expression

$$\boxed{\frac{dJ(u_\theta;x_s,s)}{d\theta} = \int_s^1 \frac{\partial}{\partial\theta}f\big(u_\theta(X_t^u,t),t\big)\,dt + \int_s^1 \frac{\partial u_\theta(X_t^u,t)}{\partial\theta}\,\sigma(t)\,\nabla_{x_t}J(u_\theta;x_t,t)\,dt} \quad (20)$$

## B    LATENT SPACE INTERPOLATABILITY

To analyze whether the one-step model learned by our approach captures an interpolatable latent space, we examine the model's outputs when fed with interpolated noise samples between two independently drawn Gaussian noise vectors. Specifically, given $x_0, x_0' \sim \mathcal{N}(0,I)$, we generate interpolated inputs using

$$x_\alpha = \sqrt{1-\alpha}\,x_0 + \sqrt{\alpha}\,x_0', \quad \alpha \in [0,1].$$

We then apply one-step denoising to each interpolated sample using our shortcut-CSL model. The resulting images shown in Figure 4 exhibit smooth transitions across the interpolation path, suggesting that the model has an interpolatable latent space.

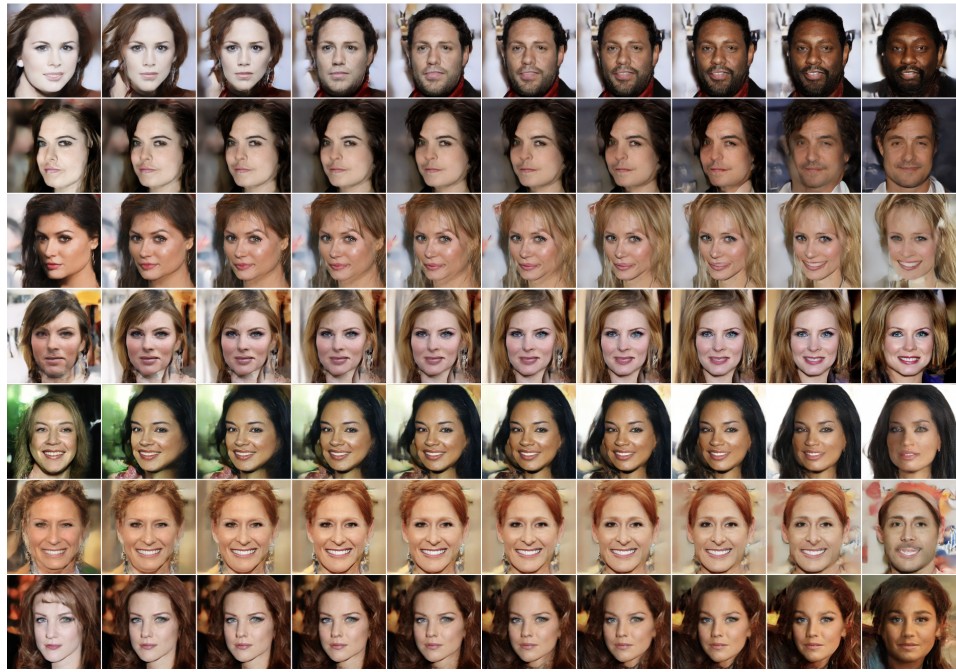

Figure 4: Image generated using the shortcut-CSL (ours) method with one-step denoising applied to a variance-preserving interpolation between two Gaussian noise samples. The leftmost and rightmost images correspond to independently drawn noise samples, while the intermediate images were produced from interpolated samples.

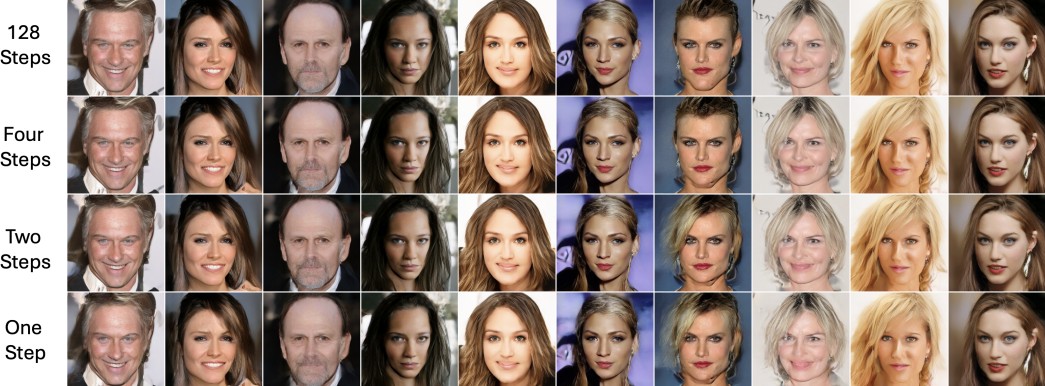

Figure 5: Image generated by the shortcut-CSL (ours) method by four-, two-, and one-step denoising. 128 steps denoising corresponds to the base flow-matching model.

## C    SAMPLES FROM SHORTCUT-CSL

In Figure 5, we provide some samples from the proposed shortcut-CSL method for different generation budgets.

## D    HYPERPARAMETER DETAILS

For experiments in Tables 1 and 6, we use the medium-scale diffusion transformer DiT-B-2 (Peebles & Xie, 2023) as the backbone network for both the shortcut baseline and our method. For ImageNet-256, due to computational limitations, we sample a subset of 30 classes with ∼40K total image samples for class-conditional training. For this and also CelebA-256, we downsample images from

`3×256×256` to `4×32×32` using the VAE encoder from the Stable Diffusion framework (Rombach et al., 2022), specifically the `sd-vae-ft-ema` variant. We then train the diffusion model in this compressed latent space. Detailed hyperparameters for our (ST-CSL) approach are provided in Table 7. Here, $M$ denotes the total number of denoising steps in the base flow-matching model. The baseline shortcut method (ST) uses the same hyperparameter configuration, except with $R = 1$. The experiments were run on NVIDIA A100 GPU.

Table 7: Hyperparameter setting for the reported results in Tables 1 and 6.

| Hyperparameter | Value |
|---|---|
| epochs | 300 (CelebA), 500 (CIFAR10, ImageNet) |
| $M$ | 128 |
| $B$ | 64 (CelebA, CIFAR10), 128 (ImageNet) |
| $K$ | 16 (CelebA, CIFAR10), 32 (ImageNet) |
| $R$ | 2 |
| ema | 0.9999 |
| optimizer | AdamW |
| learning rate | 0.0001 |
| weight decay | 0.1 |

## E  ALGORITHM

The algorithm for training shortcut models with the proposed CSL loss is detailed in Algorithm 1.

---

**Algorithm 1** Training Shortcut Models with cumulative Self-Consistency Loss (Shortcut-CSL)

---

1: $K \leftarrow$ #bootstrap targets
2: $B \leftarrow$ #flow-matching targets
3: $R \leftarrow$ #terms in $J_{CSL}$ estimation
4: **while** not converged **do**
5: $\quad x_0 \sim \mathcal{N}(0, I), \quad x_1 \sim D, \quad (d, t) \sim p(d, t)$
6: $\quad x_t \leftarrow (1 - t)x_0 + tx_1$
7: $\quad$ **for** $B$ batch elements **do**
8: $\quad\quad \mathcal{S}_{\text{target}} \leftarrow x_1 - x_0$
9: $\quad\quad d \leftarrow 0$
10: $\quad$ **end for**
11: $\quad$ **for** $K/R$ batch elements **do**
12: $\quad\quad s \leftarrow t, x_1' \leftarrow x_t, \mathcal{S}_{\text{target}} = []$
13: $\quad\quad$ **for** $R$ iterations **do**
14: $\quad\quad\quad \mathcal{S}_1 \leftarrow \mathcal{S}_\theta(x_1', s, d)$
15: $\quad\quad\quad x_2' \leftarrow x_1' + \mathcal{S}_1 d$
16: $\quad\quad\quad \mathcal{S}_2 \leftarrow \mathcal{S}_\theta(x_2', s + d, d)$
17: $\quad\quad\quad$ APPEND($\mathcal{S}_{\text{target}}$, `stopgrad`$((\mathcal{S}_1 + \mathcal{S}_2)/2)$)
18: $\quad\quad\quad x_1' \leftarrow x_1' + \mathcal{S}_\theta(x_2', s + d, 2d) \cdot 2d$
19: $\quad\quad\quad s = s + 2d$
20: $\quad\quad$ **end for**
21: $\quad$ **end for**
22: $\quad s \leftarrow t, x_1' \leftarrow x_t$
23: $\quad$ **for** $r = 1$ to $R$ **do**
24: $\quad\quad \theta \leftarrow \nabla_\theta \|\mathcal{S}_\theta(x_1', s, 2d) - \mathcal{S}_{\text{target}}[r]\|^2$
25: $\quad\quad x_1' = x_1' + \mathcal{S}_\theta(x_1', s, 2d) \cdot 2d$
26: $\quad\quad s = s + 2d$
27: $\quad$ **end for**
28: **end while**

---

The cumulative grad term in $\nabla_x J$ ( Equation (13)) term is realized in practice by line 24 of Algorithm 1. For $R = 2$, lines 23–26 operate as follows: Starting from an initial state $x'_1$ at time $t$, we compute a shortcut step with stepsize $2d$ as:,

$$\mathcal{S}_\theta(x'_1, t, 2d),$$

and evaluate the squared-loss objective against the fixed target produced by rolling out two steps with stepsize $d$ (lines 13–19). In practice, PyTorch autograd computes this gradient. We then move along the shortcut step to obtain the "next state" and updated time (line 25 and 26). Now, in the second iteration, we again compute $\mathcal{S}_\theta(x'_1, t, 2d)$ at this "next state", and evaluate the squared loss with the corresponding target. Crucially, during this second loss evaluation, the "next state" depends on the shortcut step taken in the first iteration (line 25). As a result, the gradient from the second loss flows backward into the computation of the first shortcut step. This ensures that the first shortcut step is also updated so as to reduce the loss incurred at the second step. This is precisely how the gradient backpropagates from the second (future) step to the first (current) step and realizes the cumulative gradient term in Equation (13).

## F    CONNECTION TO REINFORCEMENT LEARNING

The optimal control (OC) viewpoint of few-step models naturally connects to reinforcement learning. In OC literature, the following OC objective (Equation (2)) at optimality is known as a *value function*, which mirrors the RL value function that represents the cumulative future cost along a trajectory:

$$J(u_\theta; x_s, s) = \int_s^1 \left( f(u_\theta(X^u_t, t), t) + g(X^u_t, t) \right) dt + h(X^u_1), \qquad X^u_s = x_s$$

Since our proposed $J_{\mathrm{CSL}}$ has the same structure—accumulating the consistency error $\|u_\theta\|^2$ along the trajectory—we can formulate training as an RL problem: the shortcut model acts as the actor, while a value network serves as the critic estimating $J_{\mathrm{CSL}}$ via temporal-difference learning. We can formulate our method as a reinforcement problem as follows:

Starting with random noise at $t = 0$, an agent aims to transform it into a meaningful image in $(1/d)$-steps, taking a each step of size $d$ going from $t = 0$ to $t = 1$. The agent is trained to take a direction, moving along which generates the best image (i.e., same as the direction taken by the base 128-step model in our case). For this, we define the reward as the negative of the CSL $(-J_{CSL})$, which we aim to maximize.

$-J_{CSL}$ acts as our value function, and we can train a value network to estimate it. We can use the following temporal difference learning to learn a value network $V_\phi$:

$$V_\phi(x_t, t, d) = -\|u_\theta(x_t, t, d)\|^2 + \gamma \, V_\phi(x_{t+d}, t + d, d) \qquad (21)$$

where $x_{t+d}$ is calculated using Equation (5). Here, $u_\theta(x_t, t, d)$ is the immediate misalignment where $\theta$ is the parameter of the few-step model, and $V_\phi$ represents the future misalignment from time $t + d$ onwards, predicted by the value network.

In reinforcement learning terms, the $(1/d)$-step model serves as an *actor*, and the value model serves as a *critic*, where the actor tries to maximize the estimated value. Instead of training a separate value network to estimate CSL, we opted for $R$-rollout (especially, $R = 2$) as it offered a good balance of performance and efficiency. We leave further exploration with techniques from the reinforcement learning literature for future work.

## G    EMPIRICAL TIME AND MEMORY CONSUMPTION COMPARISON

Table 8 reports the wall-clock time (in hours) and GPU memory usage (in GB) for training Flow-Matching (FM), shortcut model (ST), and our method (ST-CSL) on the CelebA dataset for 100 epochs, using small, medium, and large Diffusion Transformer (DiT) models. All experiments were conducted on a single NVIDIA A100 GPU.

Table 8: Wall-clock time (in hours) and GPU memory usage (in GB) for training a Flow-Matching (FM), a shortcut (ST) and Ours (ST-CSL) methods for Diffusion Transformers (parameter counts) of small, medium, and large sizes.

| Model | Time (hours) | | | Memory (GB) | | |
|---|---|---|---|---|---|---|
| | FM | ST | ST-CSL | FM | ST | ST-CSL |
| DiT-S-2 (33M) | – | 5.6 | 6.2 | – | 21.35 | 21.35 |
| DiT-B-2 (130M) | 6.4 | 7.4 | 7.8 | 28.66 | 31.67 | 28.96 |
| DiT-L-2 (458M) | – | 11.4 | 12.2 | – | 45.47 | 38.52 |

FM is the most efficient of all in time and memory, but does not support few-step generation.

Our method requires slightly more time than the shortcut model but uses less memory in larger models. This trade-off arises from how each method processes batches during training. The shortcut model processes the entire batch in a single step from the current state, which minimizes time but leads to high memory consumption. In contrast, our method uses a two-step process: it first computes an intermediate step using part of the batch, then computes the next step using the intermediate outputs (as described in lines 23–26 in Algorithm 1). Splitting the batch this way reduces memory usage but requires additional time for the extra computation.

For two-stage methods like Progressive Distillation (PD) and Reflow, we first need to train the FM model and distill the knowledge to the student model. Reflow requires twice the time taken by FM. For PD, we train a student model that learns to generate samples in half as many steps as the teacher model. Therefore, a separate model needs to be trained for 128, 64, 32, ..., 2 and 1 step generation successively, consuming a significantly large amount of time.

## H  METRIC DETAILS

**Fréchet Inception Distance (FID).**    FID measures the discrepancy between the real and generated data distributions by comparing their feature-level means and covariances. The features are obtained from an Inception v3 network. Lower FID indicates that the generated distribution matches the real distribution more closely. Given real features $F^r = \{f_i^r\}_{i=1}^{N_{fr}}$ and generated features $F^g = \{f_j^g\}_{j=1}^{N_{fg}}$, let $(\mu_r, \Sigma_r)$ and $(\mu_g, \Sigma_g)$ denote their empirical means and covariances.

$$\text{FID} = \|\mu_r - \mu_g\|_2^2 + \text{Tr}\left(\Sigma_r + \Sigma_g - 2(\Sigma_r \Sigma_g)^{1/2}\right). \tag{22}$$

**Precision**    Precision measures the *quality* of generated samples by quantifying the fraction of generated points that lie within the local support of the real data manifold. For each real feature $f_i^r$, define the $k$-nearest-neighbor radius

$$\delta_i^{(r)} = \text{kNN\_dist}(f_i^r, F^r), \qquad i = 1, \ldots, N_r. \tag{23}$$

A generated point $g_j$ is considered valid if it lies within the radius of at least one real point. Precision is:

$$\text{Precision} = \frac{1}{N_g} \sum_{j=1}^{N_g} \mathbf{1}\left[\min_i \|f_j^g - f_i^r\|_2 \leq \delta_i^{(r)}\right]. \tag{24}$$

**Recall**    Recall measures the *diversity* of generated samples by quantifying how much of the real data manifold is covered by the generated distribution. For each generated feature $f_j^g$, define

$$\delta_j^{(g)} = \text{kNN\_dist}(f_j^g, F^g), \qquad j = 1, \ldots, N_g. \tag{25}$$

A real point $f_i^r$ is considered covered if a generated sample lies within its radius. Recall is:

$$\text{Recall} = \frac{1}{N_r} \sum_{i=1}^{N_r} \mathbf{1}\left[\min_j \|f_i^r - f_j^g\|_2 \leq \delta_j^{(g)}\right]. \tag{26}$$

**F1 Score.** The F1 score summarizes precision and recall into a single measure, balancing sample quality and diversity. It is defined as the harmonic mean:

$$F_1 = \frac{2 \cdot \text{Precision} \cdot \text{Recall}}{\text{Precision} + \text{Recall}}. \tag{27}$$

# I CONVERGENCE WITH TRAINING STEPS

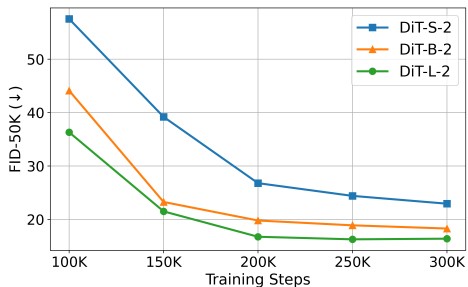

Figure 6: We evaluate the convergence of shortcut-CSL by tracking the FID score at regular training intervals across different backbone DiT network sizes on the CelebA-256 dataset. Our results show that performance consistently improves with increasing model size at every stage of training.

# J LLM USAGE

LLM is lightly used to polish writing in the paper. The used prompt is `Please polish the following sentence/sentences`

