# OpenReview forum: "Shortcut Diffusion Training with Cumulative Consistency Loss: An Optimal Control View"
_ICLR.cc/2026/Conference — ICLR 2026 Poster_

### Official Review · Reviewer_sxLT · 2025-10-27

**Soundness:** 3
**Presentation:** 3
**Contribution:** 3
**Rating:** 4
**Confidence:** 3

**Summary:**

The paper addresses the low sampling efficiency of diffusion/flow models by enabling one- or few-step generation. Building on “shortcut” samplers that learn large step sizes via a self-consistency loss against a many-step base flow-matching model, the authors reinterpret this loss through optimal control, treating few-step sampling as a controlled base process. From this view, they propose a cumulative self-consistency loss (CSL) that penalizes misalignment not only at the current step but also along future steps of the trajectory, encouraging large yet reliable steps that maintain downstream sample quality. They further draw connections to reinforcement learning. Experiments indicate improved one-/few-step generation quality under the same training budget, and the proposed shortcut-CSL variant outperforms baselines on certain tasks.

**Strengths:**

• The paper is clearly written with precise notation; figures and tables are informative and easy to follow.

• The proposed shortcut-CSL shows better performance than prior shortcut variants on some tasks, supporting the value of the cumulative alignment objective.

**Weaknesses:**

1. Metrics are too narrow. Evaluating primarily with FID is insufficient, and the observed FID gains are minor in some cases. Please broaden the evaluation metrics like F1, CLIP-score, and qualitative comparisons, and, if feasible, a small human preference study. This multifaceted assessment would make the empirical case more solid.

2. The teaser (Fig. 1) visualizations look only slightly improved over baselines.

3. The dataset scope is limited. Results are reported only on CelebA and CIFAR-10. While broader evaluation is costly, for fair comparisons with other methods, you should include additional datasets—e.g., ImageNet (64×64 and/or 256×256), FFHQ, or another modern benchmark—while keeping ablations on a single dataset to manage workload. This would better test generality and strengthen claims about scalability and robustness.

**Questions:**

My primary concern is the limited practical performance gains demonstrated (see Weaknesses).

---

> ### Author Response · Authors · 2025-11-21
> **Response to the Review**
>
> We want to thank Reviewer sxLT for taking the time to review our work and providing feedback for improvement. The following are our responses to the concerns:
>
> **My primary concern is the limited practical performance gains demonstrated.**
>
> $\rightarrow$
>
> The best performance for shortcut models (ST) and our (ST-CSL) methods is at the ratio B: K = 1:1, as reported in Table 4 in the paper. For CelebA, the FID for 128-step generation is 8.01, while two-step generation with ST is 11.14. ST-CSL reduces the two-step FID to 9.91, narrowing the gap to within 2 points of the 128-step model. For CIFAR-10, ST-CSL method achieves an FID of 7.94 for two-step generation (which is within 1.5 points of the 128-step generation of ST), compared to 10.54 for the ST model. On average, our method's performance in this setting reaches within 24% of the 128-step generation, compared to 42% for ST—a significant relative gain. These results demonstrate that our method significantly improves two-step generation quality, bringing it much closer to the 128-step performance.
>
> To further demonstrate significant performance gain in a challenging dataset, we experimented on a class-conditional generation task on ImageNet-256, and results show significantly improved results in terms of FID and F1 score, which we have detailed in the subsequent comments.
>
> ---
>
> **The dataset scope is limited. Results are reported only on CelebA and CIFAR-10. Please broaden the evaluation metrics**
>
> $\rightarrow$
>
> We have conducted additional experiment on the class-conditional generation task on ImageNet-256, and the results are reported in the following table. We calculate the FID and the F1 score of the generated samples, the additional metric F1-score jointly measures the sample quality and diversity compared to the original data samples (the details of these metrics are in Appendix H).
>
> | **Method** | **Four-Step (FID↓)** | **Two-Step (FID↓)** | **One-Step (FID↓)** | **Four-Step (F1↑)** | **Two-Step (F1↑)** | **One-Step (F1↑)** |
> |:-------:|:-----:|:------:|:------:|:------:|:-----:|:-----:|
> | **Reflow** | 40.86 | 43.68 | 50.40 | 0.56 | 0.54 | 0.51 |
> | **FM** | 104.96 | 210.39 | 325.78 | 0.26 | 0.09 | 0.00 |
> | **Meanflow (15%)** | 48.22 | 50.13 | 59.63 | 0.55 | 0.55 | 0.54 |
> | **Meanflow (10%)** | 39.65 | 41.10 | 50.01 | 0.57 | 0.57 | 0.55 |
> | **Meanflow (5%)** | 34.09 | 36.61 | 45.12 | 0.58 | 0.57 | **0.56** |
> | **ST (4:1)** | 24.70 | 35.73 | 64.12 | 0.61 | 0.56 | 0.46 |
> | **ST (2:1)** | 23.47 | 32.54 | 55.55 | 0.63 | 0.58 | 0.50 |
> | **ST (1:1)** | 24.17 | 32.22 | 51.78 | 0.63 | 0.59 | 0.51 |
> | **ST-CSL (4:1)** | 16.98 | 21.77 | 45.84 | 0.63 | 0.60 | 0.50 |
> | **ST-CSL (2:1)** | 16.21 | 18.71 | 37.60 | 0.64 | 0.62 | 0.53 |
> | **ST-CSL (1:1)** | **15.71** | **17.35** | **31.66** | **0.64** | **0.63** | **0.56** |
>
> Shortcut(ST) and our (ST-CSL) are trained at two different ratios $B:K$. The results show significant gains when using CSL over SL, and the gain is even more significant in advanced and challenging datasets like ImageNet-256. The FID for base 128-step generation is 15.21. For 4-step generation, ST-CSL achieves similar performance as the 128-step generation, and the FID for 2-step is within 2.5 points of the 128-step model. This is a significant improvement compared to the ST.
>
> The table and all details are added to section 6.5 of the manuscript.
>
> The time consumed (in hours) per 100 epochs of training is reported in the table below:
> | Model             | **FM** | **ST (2:1)** | **ST-CSL(2:1)** |
> |-------------------|-------:|-------:|-----------:|
> | DiT-B-2 (296M)    |   5.6  |   6.5  |     7.5    |
>
> We will add experiments for class-conditional ImageNet-256 to other baselines (present in Table 1 in the manuscript) as well for the camera-ready version.
>
> ----------------------
>
> **The teaser (Fig. 1) visualizations look only slightly improved over baselines.**
>
> $\rightarrow$
>
> The visualization is to demonstrate that shortcut models tend to develop artifacts when generating with one or two steps, and that our method improves upon that with cleaner images.
>
> We have updated Figure 1 with generation on the ImageNet-256 dataset. Images produced by ST with one or two steps exhibit noticeable artifacts; for instance, in the flying-bird example (in the figure), despite the plain background, distinct distortions appear around the object. In contrast, our method yields smoother and cleaner generations. This is also supported by the values of the precision metric, which quantifies the quality of the generated images: the precision for ST-CSL (2:1) generated samples for two and one-step generations are 0.60 and 0.49, but that of the ST (2:1) is 0.52 and 0.42.
>
> ------------------------------------------------------------
>
> We hope the result for ImageNet, showing a near 20 points improvement in FID and 4% improvement in F1-score for one-step generation, addresses your primary concern of limited gains with CSL. Please let us know if you have further comments.

---

> > ### Comment · Reviewer_sxLT · 2025-11-26
> >
> > Thank you for your rebuttal, which address my concerns, and I would like to raise my point.

---

> > > ### Author Response · Authors · 2025-11-26
> > > **Thank you!**
> > >
> > > Dear Reviewer sxLT,
> > >
> > > Thank you for the discussion and for supporting our research. The feedback has strengthened our empirical analysis and helped us better demonstrate the advantages of our proposed method.

---

> > > > ### Author Response · Authors · 2025-12-03
> > > > **Update: Addition of the recent "Meanflow" baseline for comparison**
> > > >
> > > > To further strengthen the empirical analysis and better frame our empirical performance, we have also included the recent Meanflow [1] method as a baseline for comparison.
> > > >
> > > > The table in the previous comment has been updated with the latest Meanflow results. The results show significant gains of our method (ST-CSL) when compared to Meanflow and ST. Additionally, although Meanflow performs better than ST in one-step generation, ST offers better performance vs efficiency tradeoff as shown by its superior performance at two- and four-step generations.
> > > >
> > > > **We have kept these results in section 6.5 of the paper.**
> > > >
> > > > [1] Geng et al., Mean Flows for One-step Generative Modeling. NeurIPS 2025.

---

### Official Review · Reviewer_CsHB · 2025-10-28

**Soundness:** 3
**Presentation:** 3
**Contribution:** 3
**Rating:** 4
**Confidence:** 3

**Summary:**

This paper improves upon the original shortcut model paper by introducing a cumulative self-consistency loss. In the paper, the original shortcut model loss is first framed as an optimal control problem, where the objective is to minimize *total* accumulated error. However, the standard loss only penalizes error at the *current* timestep, and thus does not account for future error. This paper derives the optimal objective that maximizes for future error minimization, analogous to maximizing future return rather than reward in an RL framework. Practically, it is found that using a discrete number of future steps (R=2 or R=4) already leads to sizable improvement. Experiments show that the proposed cumulative SL reliably improves upon the base SL, and ablations show that the improvement can be steadily improved with additional compute allocated to R.

**Strengths:**

This paper highlights a well-posed analysis of the shortcut modelling loss, showing that is an instantaneous approximation to an objective that should be expanded to account for future errors. Using this viewpoint, the paper derives a clear true objective, and shows that even a discrete approximation to this true objective results in performance improvement. In this way, the paper clearly ties together empirical improvement towards a theoretically-motivated insight.

The clarity in the paper is passable. Figure 2 provides a solid intuition as to why the cumulative objective is a more precise target. (See below for weaknesses).

The significance of this paper is solid, as it provides a new viewpoint which can be applied to distillation methods in general. The experiments are conducted on standard benchmarks at a reasonable network size, and wall-clock experiments are presented to take computational requirements into account. Error bars are included.

**Weaknesses:**

The empirical performance of this method could be better framed by comparing to recent prior works, specifically Meanflow.

In terms of clarity, the notation introduced in Section 3/4/5 can be refined. A full explanation of optimal control as given in section 3 may not be necessary, especially terms such as b, g, and h which are not used in the final relation to shortcut model loss. In equation 4, it will help to make it clear that u(X) is the *error* term which has an optimality at zero, but the network itself does not predict u(X) but rather a velocity (that must be compared to the teacher velocity).

For equation 14, is the gradient with relation to future errors taken through multiple evaluations of the network? If so, is the error term for k>1 backpropgated to all previous steps including the current step? An algorithmic explanation would strengthen this section.

While the text argues that equations 11 and 12 should be different, the resulting integrals appear identical. What is the precise difference between these two objectives? Commentary would strengthen this section.

**Questions:**

See section above for further questions associated with each topic.

As a side note, the discussion in the Appendix on the connection to TD-based value learning methods is quite interesting, and could be a worthy direction for future work.

In general, the ideas in this paper can be strengthened by a more thorough clarity in writing. The introduction and background sections can be condensed, and the relation between optimal control and the shortcut loss can benefit from additional textual commentary explaining the intuition behind the equations. With clarifications to the questions in the weaknesses section, I would consider a raise to the review score.

Can the cumulative self-consistency objective be applied to other few-step techniques such as consistency models or meanflow?

---

> ### Author Response · Authors · 2025-11-21
> **Response to the Review**
>
> We want to thank Reviewer CsHB for recognizing our contribution and raising important points regarding the clarity of the paper. Accordingly, we have revised Sections 3.1, 4, 5, and Appendix E to significantly improve the clarity of our methodological details. Also, we added Section 6.5 with an experiment on class-conditional generation on a more challenging ImageNet-256 dataset, where our method demonstrates even greater improvement over the baseline shortcut model. All changes are highlighted in blue.
>
> Below are our responses.
>
> **The relation between optimal control and the shortcut loss can benefit from additional textual commentary explaining the intuition behind the equations.**
>
> $\rightarrow$
>
> The following equations formulate the shortcut model as an optimal control problem (Equations 8 and 9):
>
> Eq:8 | $dX^u_t = \big[b(X^u_t, t) + u_\theta(X^u_t, t)\big]\, dt$
>
> Eq:9 | $J(u_\theta; x_{t'}, t') = \int_{t'}^1 f\big(u_\theta(X_t^u, t), t\big)\ dt$
>
> - Equations 8 and 9 reveal that changing the model parameters $\theta$ affects the cumulative objective $J$ in two ways: (i) by altering the immediate error $u_\theta(X^u_{t'}, t')$, and (ii) by changing the subsequent states $X^u_t$, which in turn impacts the future errors $u_\theta(X^u_t, t)$ for $t > t'$. We have shown in Lemma 1 that the self-consistency loss accounts only for the effect in the objective $J$ due to the immediate error, discarding the downstream effects brought by the changes in the subsequent states $X^u_t$
>
> - Technically, Equation 9 accumulates the cost $f$ over the full trajectory from $t = t'$ to $t = 1$. Lemma 1 shows that, SL loss corresponds to restricting this accumulation to only the current time via a Dirac delta function. Thus, $J_{SL}$ is suboptimal: it penalizes the error $u_\theta$ only at $t'$, ignoring how this error propagates and affects future errors at $t>t'$ along the trajectory
>
> We have included this in the manuscript.
>
> -----
>
> **While the text argues that equations 11 and 12 should be different, the resulting integrals appear identical. What is the precise difference between these two objectives? Commentary would strengthen this section.**
>
> $\rightarrow$
>
> We had a subtle error in the equation, which we have corrected. The functional form of $f$ for USL and CSL are as follows:
>
> Eq:11 (USL) | $f\big(u_\theta(X_{t}^u, t),t \big) = \| u_\theta(x_{t'}, t')\|^2\,\ \forall t>t'$, where, $x_{t'}=X_{t'}^u$
>
> Eq:12 (CSL) | $f\big(u_\theta(X_t^u, t),t \big) = \| u_\theta(X_t^u, t)\|^2\ , \forall t>t'$
>
> - $J_{USL}$ relaxes the delta function constraint and considers errors over the entire time $t'<t<1$; however, it has a naive assumption that the error $u_\theta(X_{t}^u, t)$ is independent of $t$, and is uniform over the entire trajectory which is equal to the value at $t=t'$. Essentially, this simply reduces to a time-weighted SL loss.
> - CSL loss further relaxes the uniformity assumption and captures the true error landscape along the entire trajectory, aggregating the error at each time step.
>
> To visually clarify the difference between the objectives, we have added a subfigure i.e., Figure 2(b). And added a commentary about the difference in the losses towards the end of section 5. All the added text is highlighted in blue.
>
> --------
>
> **For equation 14, is the gradient with relation to future errors taken through multiple evaluations of the network? If so, is the error term for $k>1$ backpropagated to all previous steps, including the current step? An algorithmic explanation would strengthen this section.**
>
> $\rightarrow$
>
> Here, we present a walkthrough of the gradient computation steps presented in Algorithm 1 (Appendix E). The gradient term is realized in practice by line 24 of Algorithm 1. For $R = 2$, lines 23–26 operate as follows:
> -  Starting from an initial state $x'\_t$ at time $t$, we compute a shortcut step with stepsize $2d$ as  $s_\theta(x'_t, t, 2d)$, and compute the gradient of the squared-loss objective against the fixed target produced by rolling out two steps with stepsize $d$. model (lines 13–19). In practice, PyTorch autograd computes this gradient.
>
> - We then move along the shortcut path to obtain the "next state" and updated time (line 25 and 26). Now, in the second iteration, we again compute $s_\theta(x'_t, t, 2d)$ at this "next state", and compute the gradient of the squared loss with the corresponding target.
>
> - Crucially, during this second gradient computation, the “next state" depends on the shortcut step taken in line 25 of the previous iteration. As a result, the gradient from this second step also flows backward into the computation of the first shortcut step. This ensures that, in addition to the second step, the first shortcut step is also updated so as to reduce the loss incurred at this step. This is precisely how the gradient backpropagates from the second step to the first step, which corresponds to the "cumulative gradient" term in Equation 13.
>
> We have added these details in Appendix E.

---

> ### Author Response · Authors · 2025-11-21
> **Response Contd...**
>
> **In terms of clarity, the notation introduced in Section 3/4/5 can be refined. A full explanation of optimal control as given in section 3 may not be necessary, especially terms such as b, g, and h, which are not used in the final relation to shortcut model loss.**
>
> $\rightarrow$
>
> Since we get to use an added page during rebuttal and the final version, we believe that with the increased page limit, we can fit the background on optimal control for better context. However, we cut out the mathematical equation and added some interpretation of the gradient in section 3.1.
>
> - The gradient in Equation 3 consists of an immediate and a propagated effect. The first term measures how changes control parameter θ directly influence the instantaneous control cost, while the second term captures how those same changes alter the future trajectory and thus the future costs. Together, they quantify both the local and downstream impacts of the change in θ.
> - In section 4, we have included the following discussion on why terms g and h, that appear in optimal control literature are omitted in our objective in Equation 9: since our goal is solely to make the shortcut model's trajectory match the base model, we focus only on minimizing the error $u_\theta$ and do not consider intermediate state costs $g(X_t^u, t)$ or terminal state cost $h(X_1^u)$. Therefore, we set $g=h=0$ and redefine the optimal control objective in Equation 9.
>
> -----
>
> **In equation 4, it will help to make it clear that u(X) is the error term which has an optimality at zero, but the network itself does not predict u(X) but rather a velocity (that must be compared to the teacher velocity).**
>
> $\rightarrow$
> We have included this detail in section 4 where we formulate the shortcut model from a control-theoretic viewpoint.
>
> ----
>
> **The empirical performance of this method could be better framed by comparing to recent prior works, specifically Meanflow.**
>
> $\rightarrow$
>
> Meanflow (Geng et al., NeurIPS 2025) is published very recently, less than a week before our submission to ICLR, and unfortunately, we were not aware of the paper at that time.
>
> Our primary contribution lies in analyzing and improving the self-consistency loss, making the Shortcut model our most meaningful baseline for comparison, which we have done extensively. Since other baselines focus on a different training paradigm, our theoretical framework might be less suitable for them. Therefore, we focused less on those works as a comparison with them would not warrant support for our contribution.
>
> -----
>
> **Can the cumulative self-consistency objective be applied to other few-step techniques, such as consistency models or meanflow?**
>
> $\rightarrow$
>
> Our method addresses the suboptimality in the self-consistency loss, which arises from the following generative equation where a sample is obtained by iteratively solving it ($s$ is the shortcut model):
> $x_{t+d} = x_t + s(x_t, t, d) \cdot d$
>
> We improve the self-consistency loss by considering not only the immediate misalignment but also future misalignments along the generation trajectory.
>
> - **Consistency models** have a different training paradigm. The following is the generative equation for consistency models, where a sample (at $t = 1$) is obtained in a single step:
>
>     $x_1 = F_\theta(x_0,0)$
>
>     The multi-step version involves perturbing $x_1$ again by adding some noise and denoising that again to obtain a refined sample $x_1$ (Song et al, ICML 2023; Algorithm 1):
>
>     $x_{\tau} = x_1 + {(\tau^2} - \epsilon^{2})z\ , \ z \sim N(0,1)$
>
>     $x_1 \leftarrow f_\theta(x_\tau,\, \tau)$
>
>     Since a consistency model learns a direct mapping from noisy samples at intermediate time to the data sample at final time ($t=1$), it cannot be relaized as a controlled process like shortcut models. For such single-step mapping, there are no future steps and thus no future misalignments. This makes our framework not applicable to its training.
>
> - Regarding **Meanflow**, it generalizes consistency models by learning a direct mapping from a noisy sample at any time $r$ to any future time $t > r$ (with consistency models using the special case $t = 1$). This makes generation a gradual denoising process along a trajectory $t: 0 \rightarrow t_1 \rightarrow t_2 \rightarrow \cdots \rightarrow 1$ similar to shortcut models. However, its training loss differs from the self-consistency loss: it includes gradient-dependent terms (Geng et al., NeurIPS 2025, Eq. 9). Thus, the training objective cannot be interpreted as solving an optimal control problem. That said, we believe a notion of cumulative loss along the generation trajectory could still be formulated, and we leave this exploration to future work.
>
> ----
>
> We hope our response and revision address your concerns regarding the clarity of the paper. We are happy to continue the discussion if there are ways in which the presentation can be further improved. Also, please let us know if you have additional comments.

---

> > ### Author Response · Authors · 2025-11-27
> > **Follow-Up on Discussion**
> >
> > Dear Reviewer CsHB,
> >
> > As we approach the end of the discussion period, we wanted to follow up to see whether our responses have sufficiently addressed your concerns and to ask if you would consider revising your evaluation accordingly.
> >
> > Please take a look at the updated manuscript, revised to incorporate your feedback. If you have any remaining concerns about presentation clarity, please let us know — we are happy to continue improving the paper.

---

> ### Author Response · Authors · 2025-12-03
> **Update: Addition of the recent "Meanflow" baseline for comparison**
>
> As per your suggestion to include the recent Meanflow [1] method as a baseline to frame our empirical performance better, we have compared the performance of our method with Meanflow in the class-conditional generation task on the ImageNet-256 dataset. **Section 6.5 of the manuscript is revised to include these results.**
>
> In the table, we compare the performance across different ratios $B : K$ for ST and ST-CSL, and different percentages of bootstrap targets for Meanflow. The results show significant gains of our method (ST-CSL) when compared to Meanflow and ST. Additionally, although Meanflow performs better than ST in one-step generation, ST offers better performance vs efficiency tradeoff as shown by its superior performance at two- and four-step generations.
>
> With this, we hope all your concerns are addressed.
>
> [1] Geng et al., Mean Flows for One-step Generative Modeling. NeurIPS 2025.

---

### Official Review · Reviewer_z676 · 2025-10-31

**Soundness:** 3
**Presentation:** 3
**Contribution:** 3
**Rating:** 6
**Confidence:** 3

**Summary:**

This paper proposes Cumulative Self-Consistency Loss (CSL) as an extension of the shortcut training loss (SL) used in one-step and few-step diffusion models. The authors reinterpret shortcut training through an optimal-control lens. CSL generalizes this by integrating consistency along the remaining trajectory, encouraging alignment not just locally but cumulatively over time. The paper derives a continuous formulation of this loss, motivates it with a “cumulative gradient” analysis, and implements a discrete estimator involving R rollouts (typically R=2). Empirically, CSL yields improved FID scores on CIFAR-10 and CelebA-256 for 1-, 2-, and 4-step sampling, with minimal additional compute cost.

**Strengths:**

- Conceptually clear reinterpretation of shortcut loss using an optimal-control framework, identifying why standard self-consistency only optimizes instantaneous alignment.
- The cumulative loss is simple to implement and shows consistent, measurable FID improvements with small computational overhead (~5–10%).
- Thorough experimental evaluation with ablations (number of rollout terms, backbone scale, training cost) and comparisons across DiT backbones.

**Weaknesses:**

- The “cumulative gradient” claimed in Eq. 13 does not seem to match the implementation. Algorithm 1 detaches rollout targets with `stopgrad` and does not backpropagate through future steps, so it seems that the optimization reduces to multiple local MSEs rather than true cumulative assignment.
- The midpoint update rule used during 2d rollouts is unconventional and unexplained.
- Evaluation scope is a bit narrow, authos only evaluate on CIFAR-10 and CelebA-256, with large baselines borrowed from prior work under different setups.

**Questions:**

1. How is the ∇ₓJ term in Eq. 13 realized in practice? Are gradients ever propagated through rollout steps, or is CSL purely a multi-point supervision objective?
2. Why does Algorithm 1 advance the 2d step from the midpoint state rather than from the starting point? Was this empirically motivated?

---

> ### Author Response · Authors · 2025-11-21
> **Response to the Review**
>
> We want to thank Reviewer z676 for the feedback and the points raised.
>
>  **The “cumulative gradient” claimed in Eq. 13 does not seem to match the implementation. Algorithm 1 (Appendix E) detaches rollout targets with *stopgrad* and does not backpropagate through future steps, so it seems that the optimization reduces to multiple local MSEs rather than true cumulative assignment.**
>
> $\rightarrow$
>
> - The use of stopgrad in line 17 of Algorithm 1 is intentional: only the targets produced by rolling out the model with step size $d$ are detached. These targets must be treated as fixed because the model with step size $2d$ is trained to follow the trajectory of the model with step size $d$, not the other way around.
>
> - The rollout with step size $2d$ is never detached with stopgrad. In lines 24–25, there is no stopgrad operation; this rollout is precisely where gradients from future steps backpropagate into earlier steps as cumulative gradients. Thus, the cumulative gradient flows entirely through the rollout of the model with step size $2d$, while the targets remain fixed.
>
> - For $R = 2$, we take two consecutive steps with step size $2d$. At each step, we compare the model’s prediction to the fixed target obtained using step size $d$. This forces the larger-step model to match the trajectory defined by the smaller-step model. If we did not stop the gradient on the targets, the smaller-step model would also receive gradients nudging it toward the trajectory of the larger-step model, which is undesirable because the smaller-step model is more accurate. Supervision must consistently propagate from smaller-step models to larger-step models.
>
> To sum up, while learning a model with step size $2d$, we obtain targets from model with stepsize $d$, and detach these targets with 'stopgrad' to treat them as fixed (line 17, Algorithm 1). Crucially, when we rollout the $2d$-step model (line 23-26, precisely line 25), we do not detach the targets (there is no stopgrad operation) to allow backpropagation of cumulative gradients. We have added this clarification along with the algorithm in Appendix E.
>
> ----------------------------------------------------------------
>
> **How is the $\nabla_x J$ term in Eq. 13 realized in practice? Are gradients ever propagated through rollout steps, or is CSL purely a multi-point supervision objective?**
>
> $\rightarrow$
>
> Adding to the previous point, the $\nabla_x J$ term is realized in practice by line 24 of Algorithm 1. For $R = 2$, lines 23–26 operate as follows:
> -  Starting from an initial state $x'\_t$ at time $t$, we compute a shortcut step with $2d$ stepsize $s_\theta(x'_t, t, 2d)$ and compute the gradient of squared-loss objective against the fixed target produced by rolling out two steps with stepsize $d$. model (lines 13–19). In practice, PyTorch autograd computes this gradient.
>
> - We then move along the shortcut path to obtain the "next state" and updated time (line 25 and 26). Now, in the second iteration, we again compute $s_\theta(x'_t, t, 2d)$ at this "next state", and compute the gradient of the squared loss with the corresponding target.
>
> - Crucially, during this second gradient computation, the “next state" depends on the shortcut step taken in line 25 of the previous iteration. As a result, the gradient from this second step also flows backward into the computation of the first shortcut step. This ensures that in addition to the second step, the first shortcut step is also updated so as to reduce the loss incurred at this step. This is precisely how the gradient backpropagates from the second step to the first step, which corresponds to the "cumulative gradient" term in Equation 13.
>
> We have added these details in Appendix E.
>
> ---------------------------
>
> **Why does Algorithm 1 advance the $2d$-step from the midpoint state rather than from the starting point? Was this empirically motivated?**
>
> $\rightarrow$
>
> We are not sure if we properly understand the Reviewer's concern. We think the Reviewer is asking why we randomly sample time $t$ (line 5 of Algorithm 1) rather than always training from the initial state at time $t=0$.
>
> A shortcut model is parametrized by $s_\theta(X_t, t, d)$, and  randomly sampling time $t$ for training ensures that shortcut models are trained to learn shortcut paths at each time step $t$. Practically for $R=2$, always starting at time $t=0$ would only train a model at $t=0$ and $t=d$, where $d$ is the step size, and the model would not learn shortcut paths at time $t>d$. Therefore, we start our rollout from a randomly sampled time-step, rather than always starting from $t=0$.
>
> --------------------------

---

> ### Author Response · Authors · 2025-11-21
> **Response contd..**
>
> **Evaluation scope is a bit narrow, authors only evaluate on CIFAR-10 and CelebA-256, with large baselines borrowed from prior work under different setups.**
>
> $\rightarrow$
>
> We have conducted additional experiment on the class-conditional generation task on ImageNet-256, and the results are reported in the following table:
>
> | **Method** | **Four-Step (FID↓)** | **Two-Step (FID↓)** | **One-Step (FID↓)** | **Four-Step (F1↑)** | **Two-Step (F1↑)** | **One-Step (F1↑)** |
> |:-----------:|:-----------------:|:-----------------:|:-----------------:|:-----------------:|:-----------------:|:-----------------:|
> | **Reflow** | 40.86 | 43.68 | 50.40 | 0.56 | 0.54 | 0.51 |
> | **FM** | 104.96 | 210.39 | 325.78 | 0.26 | 0.09 | 0.00 |
> | **Meanflow (15%)** | 48.22 | 50.13 | 59.63 | 0.55 | 0.55 | 0.54 |
> | **Meanflow (10%)** | 39.65 | 41.10 | 50.01 | 0.57 | 0.57 | 0.55 |
> | **Meanflow (5%)** | 34.09 | 36.61 | 45.12 | 0.58 | 0.57 | **0.56** |
> | **ST (4:1)** | 24.70 | 35.73 | 64.12 | 0.61 | 0.56 | 0.46 |
> | **ST (2:1)** | 23.47 | 32.54 | 55.55 | 0.63 | 0.58 | 0.50 |
> | **ST (1:1)** | 24.17 | 32.22 | 51.78 | 0.63 | 0.59 | 0.51 |
> | **ST-CSL (4:1)** | 16.98 | 21.77 | 45.84 | 0.63 | 0.60 | 0.50 |
> | **ST-CSL (2:1)** | 16.21 | 18.71 | 37.60 | 0.64 | 0.62 | 0.53 |
> | **ST-CSL (1:1)** | **15.71** | **17.35** | **31.66** | **0.64** | **0.63** | **0.56** |
>
> - Shortcut (ST) and our (ST-CSL) are trained at two different ratios $B:K$. We calculate the FID and the F1 scores of the generated samples (details of the metrics are provided in the Appendix). The FID for base 128-step generation is 15.21. The results show significant gains when using our method (ST-CSL) over ST, and the gain is even more significant in advanced and challenging datasets like ImageNet-256. For 4-step generation, ST-CSL achieves similar performance as the 128-step generation, and the FID for 2-step is within 2.5 points of the 128-step model, which is a significant improvement compared to the ST. For one-step generation, our method improves FID by 20 points and F1 score by 4% compared to ST.
>
> - Regarding baselines, please note that our primary contribution lies in analyzing and improving the self-consistency loss, making the Shortcut model our most meaningful baseline for comparison, which we have done extensively. Since other baselines focus on a different training paradigm, our theoretical framework might be less suitable for them. Therefore, we focused less on those works as comparison with them would not warrant support to our contribution.
>
>
> We have added these results in the manuscript. Please let us know if there are further comments.

---

> > ### Author Response · Authors · 2025-11-27
> > **Follow-Up on Discussion**
> >
> > Dear Reviewer z676,
> >
> > As we approach the end of the discussion period, we wanted to follow up to see whether our responses have sufficiently addressed your concerns and to ask if you would consider revising your evaluation accordingly. Please let us know if you have any additional comments.

---

> ### Author Response · Authors · 2025-12-03
> **Update: Addition of the recent "Meanflow" baseline for comparison**
>
> To further address the concern regarding our choice of baselines, we have expanded our experiment on the class-conditional generation task on ImageNet-256 to include the recent Meanflow [1] method as a baseline for comparison.
>
> The table in the previous comment has been updated with the latest Meanflow results. The results show significant gains of our method (ST-CSL) when compared to Meanflow and ST. Additionally, although Meanflow performs better than ST in one step generation, ST offers better performance vs efficiency tradeoff as shown by its superior performance at two- and four-step generations.
>
> **We have kept these results in section 6.5 of the paper.**
>
> [1] Geng et al., Mean Flows for One-step Generative Modeling. NeurIPS 2025.

---

### Official Review · Reviewer_kzb9 · 2025-11-01

**Soundness:** 3
**Presentation:** 2
**Contribution:** 2
**Rating:** 6
**Confidence:** 2

**Summary:**

The paper proposes an improvement over shortcut models to distill multi-step diffusion pipelines, thereby improving inference speed. The central idea of the paper is that the shortcut models only consider immediate errors, which can lead to misalignment with the model's true trajectory in severe step compression. The authors propose to penalise not just the immediate misalignment but also further misalignments down the trajectory of the teacher model.

**Strengths:**

- By framing the shortcut problem as an optimal control one, the authors provide mathematical grounding for their work. They model self consistency as an optimal control objective and then relax constraints to incorporate errors further along the ODE trajectory

- The authors include fair comparisons that ensures equal training budgets with baselines

- Training algorithm is included to aid reproducibility

**Weaknesses:**

- The paper lacks sufficient clarity, specifically in the connection of the work to Reinforcement Learning. Linking J_csl as a value function can be better explained, given the authors have mentioned this in the abstract as an important section of the paper.

**Questions:**

The authors discuss the increasing training cost with increasing R. Would be good to know if the authors experimented with R>4 or if the authors have an idea of how training scales with R.

---

> ### Author Response · Authors · 2025-11-21
> **Response to the Review**
>
> We want to thank Reviewer kzb9 for taking time to review our work. Following are our responses:
>
> **The paper lacks sufficient clarity, specifically in the connection of the work to Reinforcement Learning. Linking $J_{CSL}$ as a value function can be better explained, given the authors have mentioned this in the abstract as an important section of the paper**
>
> $\rightarrow$
>
> Regarding the connection between our method and reinforcement learning (RL), we have provided a detailed formulation in Appendix F.
>
> - Firstly, although the shortcut model is simple and effective, it lacks a theoretical ground. We provide a theoretical foundation by interpreting the shortcut models as a controlled process, we show that training the model corresponds to solving an optimal control (OC) problem. This theoretical grounding allows us to identify and analyze the inherent limitation of the self-consistency loss and motivates our proposed cumulated self-consistency loss.
>
> - The OC viewpoint naturally connects to reinforcement learning. In OC literature, the following OC objective (Equation 2) at optimality is known as a **value function**, which mirrors the RL value function that represents the cumulative future cost along a trajectory:
>
>   $J(u_\theta; x_{t'}, t') = \int_{t'}^1 \left(f(u_\theta(X_t^u, t), t)+ g(X_t^u, t)\right) dt+ h(X_1^u),\qquad X_{t'}^u = x_{t'}$
>
>   Since our proposed $J_{\mathrm{CSL}}$ has the same structure—accumulating the consistency error along the trajectory—we can formulate training as an **RL problem**: the shortcut model acts as the *actor*, while a value network serves as the *critic* estimating $J_{\mathrm{CSL}}$ via temporal-difference learning. Please refer to Appendix F for further details on this actor–critic formulation.
>
> -------------------------------------------
>
> **The authors discuss the increasing training cost with increasing R. Would be good to know if the authors experimented with $R>4$ or if the authors have an idea of how training scales with R.**
>
> $\rightarrow$
>
> - We did not evaluate beyond \(R = 4\) because the training cost grows with $R$. As reported in Section 6.4, $R = 4$ incurs a 30\% increase in training time compared to the shortcut model.
>
> - Table 1 and 4 show that generation quality for four-step and higher-step generation remains close to that of the 128-step base model. Significant performance degradation occurs from the two-step setting, and $R = 2$ already covers the full trajectory required for two-step generation. Therefore, there is no need to increase it further to $R= 4 \text{ or } 8$, especially as the performance for four or more steps is already close to the 128-step baseline, and thus there will be a smaller gain compared to the additional computational cost.
>
> We have also revised the paper for better clarity of sections 4, 4.1, 5 and Appendix E (based on Reviewer CsHB's comments). Please let us know if you have further comments.

---

> > ### Comment · Reviewer_kzb9 · 2025-11-25
> > **The rebuttal addresses my concerns**
> >
> > I thank the authors for the rebuttal, it addresses my concerns.

---

> > > ### Author Response · Authors · 2025-11-26
> > > **Thank you!**
> > >
> > > Dear Reviewer kzb9,
> > >
> > > Thank you for the discussion and for supporting our research.
> > >
> > > If our responses have addressed all the concerns, we kindly ask whether you might consider revisiting the rating.

---

### Author Response · Authors · 2025-11-25

Dear Reviewers,

We want to thank you for taking the time to review our work and for providing valuable feedback. We have responded to each of the concerns of all the reviewers below.

We would appreciate your feedback on whether our responses have adequately addressed your concerns. If you have any additional comments or require further clarification, please let us know.

Thank you.

---

### Author Response · Authors · 2025-11-29
**Overview on the Author-Reviewer Discussion**

Dear AC,

Since the Author-Reviewer Discussion process was interrupted due to an identity leakage issue, here we want to give an honest overview of the discussion so far, and want to know if you feel there are any remaining concerns of the Reviewers that we need to address.

**Reviewer kzb9:**

Reviewer kzb9 gave a positive rating (rating=6) initially, recognizing our theoretical contribution. The reviewer had concerns regarding the clarity of the connection of our method to reinforcement learning. After we pointed out that the details were in Appendix F, and briefly stated the details in the comment, ***the reviewer responded, stating that the concerns were addressed, and maintained the rating***.

The reviewer also had a clarification question, which we have replied to.

━━━━━━━━━━━━━━━━━━━━━━━━━━━━━━━━━━━━━━━━━━━━━━━━━━━━━━

**Reviewer z676**

Reviewer z676 gave a positive rating (rating=6).

- The reviewer had confusion regarding the algorithmic implementation of our proposed method (Equation 13). To clear the confusion, we described our algorithm thoroughly in the comment, and also revised the algorithmic details in Appendix E.

- Another weakness pointed out by the reviewer was in our experimental evaluation, especially the limited datasets and the choice of our baselines. To address this, we experimented with a more challenging ImageNet-256 dataset and presented the results in the comments. We also clarified the rationale behind the choice of baselines used for comparison with our method.

The reviewer also had clarification questions which we have replied to. ***We did not hear anything from the reviewer since the initial review.***

━━━━━━━━━━━━━━━━━━━━━━━━━━━━━━━━━━━━━━━━━━━━━━━━━━━━━━

**Reviewer CsHB**

Reviewer CsHB gave an intial rating of 4, acknowledging the soundness of our analysis and the proposed method. ***The reviewer also mentioned to consider raising the score if we can clarify the raised concerns***.

- The main concern pointed out by the reviewer was regarding the clarity of our methodological details in sections 3/4/5 and Algorithm 1, and provided suggestions for improving the writing clarity. We revised the sections accordingly, and added details of the Algorithm 1 in Appendix E. All the changes are also detailed in the comments.

- Another concern pointed out by the Reviewer was that the empirical performance of this method could be better framed by comparing to the recent "Meanflow" method. In the comments, we mentioned that the baseline was published very recently, less than a week before our submission to ICLR, and unfortunately, we were not aware of the paper at that time.

The reviewer also had a clarification question which we have replied to.

The reviewer mentioned in the review that: ***with the clarifications, the reviewer would consider a raise to the review score.*** We still had one week left in the discussion, however due to the identity leakage issue, the reviewer was cut out from the discussion, and ***we did not hear anything from the reviewer since the initial review.***

━━━━━━━━━━━━━━━━━━━━━━━━━━━━━━━━━━━━━━━━━━━━━━━━━━━━━━

**Reviewer sxLT**

Reviewer sxLT gave a rating of 4 initially.

- The reviewer mentioned that the primary concern is the limited performance gains with our method. The reviewer also mentioned that FID metric was insufficient and suggested to broaden the evaluation metrics, and to use advanced datasets like ImageNet or FFHQ for comparison. To address this, we carried out experiment of our method and baselines on ImageNet-256, and compared them based on FID and an ***additional metric: F1-score***. Results showed 20 points improvement in FID and 4% improvement in F1-score for one-step generation.

- Another weakness was that the teaser (Fig. 1) visualizations look only slightly improved over baselines. We clarified the difference between the samples from our method and the baseline, and updated the Figure with the samples on the ImageNet dataset to better support our argument.

The reviewer commented stating that: ***our response addresses the concerns, and that the reviewer would like to raise the rating, and raised the rating to 6***. However, ***because of the identity leakage issue, the review and the updated rating was reverted to their state before the start of the discussion period***.

━━━━━━━━━━━━━━━━━━━━━━━━━━━━━━━━━━━━━━━━━━━━━━━━━━━━━━

Please have a look at the discussion at your convenience and let us know if you feel there are any remaining concerns of the Reviewers that we need to address.

---

> ### Author Response · Authors · 2025-12-03
> **Update: Addition of the recent "Meanflow" baseline for comparison**
>
> To further address the concerns raised by Reviewers **z676** and **CsHB**, we added the "Meanflow"[1] method as a baseline for comparison for the class-conditional generation on the ImageNet-256 dataset. **Section 6.5 of the manuscript has been revised to include these results.**
>
> -  Reviewer **z676** had a concern with the choice of our baseline. While we had clarified about the choices in the comments, the added comparison with the "Meanflow" method further demonstrated the performance advantage of our method over the recent state-of-the-art baseline. We have posted the discussion as a comment in the corresponding review.
>
> - Reviewer **CsHB** explicitly suggested comparing with the "Meanflow" method to better frame the empirical performance of our method. We have posted the discussion as a comment in the corresponding review.
>
>
>
> [1] Geng et al., Mean Flows for One-step Generative Modeling. NeurIPS 2025.

---

### Meta-Review · Area_Chair_CoL4 · 2026-01-07

**Summary:**

Reviewers initially raised concerns regarding clarity, experimental scope, and positioning of the proposed method. Specifically, questions were raised about the connection to reinforcement learning, potential mismatches between the method and its implementation, limited evaluations on CIFAR and CelebA datasets, missing comparisons with MeanFlow, and modest performance gains.

The authors addressed these concerns by clarifying the relationship to reinforcement learning, correcting and explaining implementation details, expanding experiments to ImageNet-256, adding direct comparisons with MeanFlow, and improving the writing and presentation. Multiple reviewers acknowledged that their concerns were resolved and indicated they would remain positive or raise their scores.

Overall, the remaining concerns are minor, and reviewers agree that the revisions sufficiently strengthen the clarity, empirical support, and positioning of the paper, informing a positive decision.

**Reviewer Concerns:**

### Addressed

- **Limited experimental scope:**
  The authors expanded the experimental evaluation by adding results on ImageNet-256 and demonstrated clear performance gains, addressing concerns about limited experiments.

### Outstanding

- **N/A**

**Reviewer Scores:**

Reviewer **kzb9** raised concerns about the clarity of the connection between the proposed approach and reinforcement learning, as highlighted by the authors. The authors provided a detailed explanation clarifying this connection, and the reviewer acknowledged that the concern was addressed and indicated they would remain positive.

Reviewer **z676** expressed concerns about a potential mismatch between the proposed method and its implementation, as well as the limited evaluation on CIFAR and CelebA datasets. The authors responded with a detailed explanation and added new experiments on ImageNet-256. As a result, this reviewer is likely to remain positive or potentially raise their score.

Reviewer **CsHB** noted issues with the writing quality and the lack of comparison with MeanFlow. The authors added experimental comparisons with MeanFlow and revised the manuscript for clarity. This response is likely to lead the reviewer to raise their score.

Reviewer **sxLT** raised concerns about limited performance gains and the narrow experimental scope restricted to CIFAR and CelebA. The authors added experiments on ImageNet and demonstrated improved performance. As acknowledged by the reviewer, they are likely to raise their score.

---

### Decision · Program_Chairs · 2026-01-26

Accept (Poster)